

# Automatic adjoint-based inversion schemes for geodynamics: Reconstructing the evolution of Earth's mantle in space and time

Sia Ghelichkhan[1], Angus Gibson[1], D. Rhodri Davies[1], Stephan C. Kramer[2], and David A. Ham[3]

[1]Research School of Earth Sciences, The Australian National University, Canberra, ACT, Australia.
[2]Department of Earth Science and Engineering, Imperial College London, London, UK.
[3]Department of Mathematics, Imperial College London, London, UK.

**Correspondence:** Sia Ghelichkhan (siavash.ghelichkhan@anu.edu.au)

**Abstract.** Reconstructing the thermo-chemical evolution of Earth's mantle and its diverse surface manifestations is a widely-recognised grand challenge for the geosciences. It requires the creation of a digital twin: a digital representation of Earth's mantle across space and time that is compatible with available observational constraints on the mantle's structure, dynamics and evolution. This has led geodynamicists to explore adjoint-based approaches that reformulate mantle convection modelling as an inverse problem, in which unknown model parameters can be optimised to fit available observational data. Whilst recent years have seen a notable increase in the use of adjoint-based methods in geodynamics, the theoretical and practical challenges of deriving, implementing and validating adjoint systems for large-scale, non-linear, time-dependent problems, such as global mantle flow, has hindered their broader use. Here, we present the Geoscientific Adjoint Optimisation Platform (G-ADOPT), an advanced computational modelling framework that overcomes these challenges for coupled, non-linear, time-dependent systems. By integrating three main components: (i) Firedrake, an automated system for the solution of partial differential equations using the finite element method; (ii) Dolfin-Adjoint, which automatically generates discrete adjoint models in a form compatible with Firedrake; and (iii) the Rapid Optimisation Library, ROL, an efficient large-scale optimisation toolkit; G-ADOPT enables the application of adjoint methods across geophysical continua, showcased herein for geodynamics. Through two sets of synthetic experiments, we demonstrate application of this framework to the initial condition problem of mantle convection, in both square and annular geometries, for both isoviscous and non-linear rheologies. We confirm the validity of the gradient computations underpinning the adjoint approach, for all cases, through second-order Taylor remainder convergence tests, and subsequently demonstrate excellent recovery of the unknown initial conditions. Moreover, we show that the framework achieves theoretical computational efficiency. Taken together, this confirms the suitability of G-ADOPT for reconstructing the evolution of Earth's mantle in space and time. The framework overcomes the significant theoretical and practical challenges of generating adjoint models, and will allow the community to move from idealised forward models to data-driven simulations that rigorously account for observational constraints and their uncertainties using an inverse approach.

## 1 Introduction

Mantle convection is the 'engine' driving our dynamic planet. It is the principal control on Earth's thermal and chemical evolution, and underpins tectonic and geological activity at Earth's surface (e.g., Davies and Richards, 1992). Through interactions



with Earth's crust, it introduces heat and fluids that contribute to the formation and concentration of ore deposits (e.g., Hoggard et al., 2020). Mantle flow also induces vertical movements of Earth's surface (so-called dynamic topography, see Davies et al., 2023, for a review), leading to regional and global changes in sea level and climate (e.g., Poore et al., 2006; Moucha et al., 2008; Cloetingh and Haq, 2015). The lithosphere, considered here to be the mantle's upper thermal boundary layer, serves as a window into the form and time-dependence of mantle convection, recorded through tectonic plate motions (e.g., Iaffaldano

and Bunge, 2015; Müller et al., 2016; Stotz et al., 2018; Wang et al., 2023). Although substantial progress has been made in reconstructing the history of plate tectonic motions (e.g., Seton et al., 2012; Gurnis et al., 2012; Müller et al., 2019; Merdith et al., 2021), the quest for a dynamic reference, revealing the force equilibria within the underlying mantle, remains ongoing. In other words, the veracity of plate motion reconstructions is not matched by an equivalent knowledge of the thermochemical structure and flow history of the underlying mantle. This is a major limitation, as it inhibits our ability to understand

fundamental processes that depend on time-dependent interactions between Earth's surface and its deep interior.

        The difficulty of inferring mantle flow into the past principally occurs because the initial state is unknown. Mantle convection is an *initial condition problem*, uniquely determined by an initial condition sometime in the past: starting from some point in time, it can be uniquely modelled by solving conservation equations for mass, momentum and energy (e.g., Ricard, 2007; Rolf and Tackley, 2011). However, the lack of knowledge on this initial condition — specifically the thermochemical state of

Earth's mantle at sometime in the past — renders reconstructions of mantle flow through conventional forward calculations intractable (Bunge et al., 2003) (Figure 1-a). Moreover, current global mantle convection models employ billions of degrees of freedom and require multiple time-steps to resolve the multi-scale dynamics of Earth's mantle (e.g., Davies and Davies, 2009; Weismüller et al., 2015; Dannberg and Gassmöller, 2018; Bauer et al., 2019, 2020). Owing to the resulting computational expense, the use of conventional geophysical inverse methods, including Monte Carlo techniques (e.g. Sambridge and

Mosegaard, 2002), are considered impractical for determining past structure, dynamics and evolution of Earth's mantle.

        The initial condition problem can be partially addressed through *sequential data assimilation* techniques. In essence, the objective of sequential data assimilation is to leverage all accessible information to improve predictions of mantle flow in space and time. Data assimilation is commonly achieved through sequential filtering (e.g., Wunsch, 1996), in which the model is advanced in time over the period in which observations exist. Whenever observations become available, the model is adjusted

or 'corrected' (e.g., Bunge and Grand, 2000; Bocher et al., 2016). The magnitude of the correction can be optimally determined using methods such as the Kalman filter (e.g., Bocher et al., 2018), with the model subsequently restarted from the updated state, and this process is repeated until all available information has been utilised (Figure 1-b). In a geodynamic context, the most commonly assimilated dataset consists of paleo-surface velocities from plate tectonic reconstruction models. In recent years, there has been a notable increase in the use of this approach (e.g., Bunge et al., 2002; Davies et al., 2012; Bower et al.,

2013; Zhong and Rudolph, 2015; Nerlich et al., 2016; Young et al., 2022; Panton et al., 2023). This can be attributed to two main factors: (i) improved confidence in the validity of plate tectonic reconstruction models, and their extension further back in time (e.g., Merdith et al., 2021; Young et al., 2022; Müller et al., 2022); and (ii) the enhanced accessibility of such models, facilitated via open-source community frameworks like the GPlates project (e.g., Gurnis et al., 2012; Müller et al., 2018).





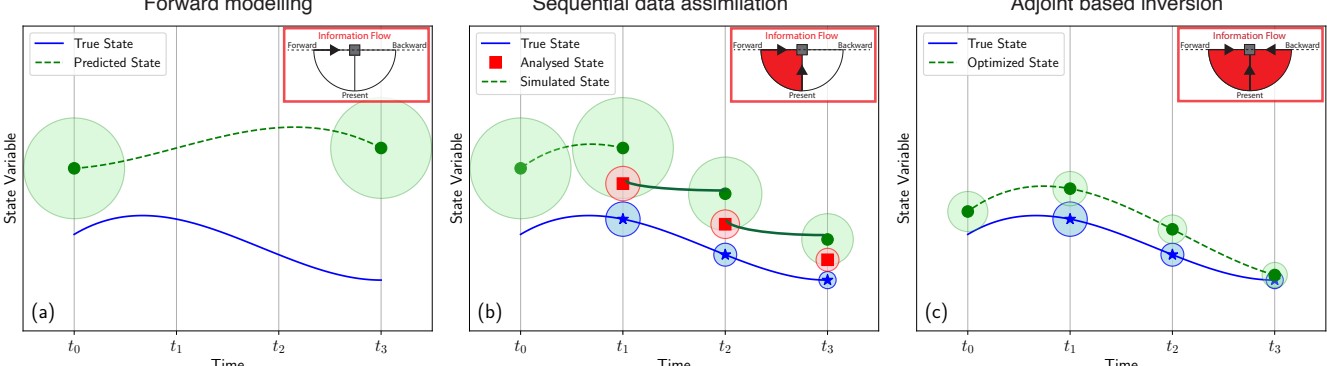

**Figure 1.** Illustration of different procedures available for estimating past mantle structure: (a) forward modelling prediction, where an unknown initial condition is estimated at $t_0$, with modelling error, measured as the difference between the modelled and true states. This difference is represented by the distance along the y-axis, which typically grows in time; (b) sequential data assimilation – starting from an initial condition at $t_0$, the forward model is run until $t_1$. An analysis is subsequently undertaken from the resulting model and the available observation. The corrected model is then subsequently integrated in time until $t_2$, when the next observation is available. This process is repeated until $t_3$ or the last time step with available observational data. The information flow diagram depicts how information is carried from both the past and present, using current data; (c) adjoint-based or variational data-assimilation, which is capable of carrying information on present-day structure (e.g., images from seismic tomography) explicitly backward in time. In (c) observational data that constrains present-day mantle structure, alongside data from different points in space and time, are used to optimise the unknown initial condition. Here, all available observations between $t_3$ and $t_0$ contribute to the analysis. The true (unknown) signal is represented by the solid line. Observations (stars), predictions (dark circles) and analyses (squares) are surrounded by ellipsoids of a size proportional to the estimated model uncertainty. Modified from Carrassi et al. (2018) and Davies et al. (2023).

Sequential data-assimilation methods, however, come with an inherent limitation: due to the sequential nature of the as-
similation process, each observation is incorporated to influence the model only at later times. Consequently, information propagates from the past into the future but cannot be transmitted back into the past. This drawback poses a significant limitation, as our knowledge of the mantle at the present day is substantially more detailed than at any other time. Thus, it becomes imperative to explore approaches that explicitly carry information backwards in time, or more precisely, enable the estimation of a time-dependent model that best fits all available observational constraints.

*Inverse geodynamics* is a rapidly emerging field that embarks on this very idea. The foundation of this field is an optimisation approach, in which mantle convection modelling is reformulated as an inverse problem. Using inverse theory, unknown model parameters can be optimised to fit available observational data, via the so-called adjoint method (e.g., Bunge et al., 2003; Ismail-Zadeh et al., 2004), through which the sensitivities of a performance measure (the so-called 'objective functional'), with respect to model parameters (e.g., the choice of initial condition) can be computed. The resulting sensitivity information can be
used to adjust model parameters, generating a model flow trajectory that matches observational constraints (e.g., present-day mantle thermochemical structure) (Figure 1-c). The geodynamic adjoint equations for reconstructing the initial condition have



been derived for isochemical, incompressible (e.g., Bunge et al., 2003; Horbach et al., 2014), compressible (Ghelichkhan and Bunge, 2016), and thermochemical mantle flow (Ghelichkhan and Bunge, 2018). Moreover, the method has been enhanced for simultaneous recovery of initial temperature conditions and rheological parameters (Li et al., 2017). Growing adoption of

adjoint-based methods within a broader geodynamic context is evidenced by their application in Reuber et al. (2020) to deci-pher subsurface structures and rheological parameters via inversion of principal stress directions, and by Crawford et al. (2018) to quantify the sensitivity of post-glacial sea level changes to lateral variations in mantle viscosity. Recent growth in computa-tional power has led to multiple adjoint-based mantle reconstructions, with a particular focus on regional geological events, as recorded in the geological record of the Americas (e.g., Spasojevic et al., 2009; Liu and Gurnis, 2010; Shephard et al., 2010)

and the Atlantic realm (e.g., Colli et al., 2018). To supplement these findings, Ghelichkhan et al. (2021) undertook a systematic global-scale comparison between adjoint model predictions and independent geological constraints, with implications for the expected rates of gravitational and dynamic ellipticity changes resulting from convection within Earth's mantle (Ghelichkhan et al., 2018, 2020).

While substantial strides have been made in the application of adjoint-based methods in geodynamics, there remain widespread

obstacles to the broader use of these techniques within the geodynamic modelling community. One key challenge is the com-plexity involved with deriving, implementing, and validating adjoint models for large-scale, non-linear, time-dependent prob-lems, core features of global mantle convection models. Owing to these difficulties, existing studies that have employed the adjoint method for reconstructing the thermochemical evolution of Earth's mantle have generally adopted a simplified ap-proach, either including a highly idealised treatment of mantle rheology (e.g., Colli et al., 2018; Ghelichkhan et al., 2021),

neglecting certain (coupling) terms in the adjoint equations, or both (e.g., Liu et al., 2008; Spasojevic et al., 2009; Liu and Gurnis, 2010; Shephard et al., 2010), which may limit the applicability of results. The work of Li et al. (2017) is a notable exception to this trend, although its focus on a 2-D rectangular domain, specifically aimed at reconstructing the dynamics of subduction, inherently limits its applicability to global mantle convection simulations. Furthermore, previous global applica-tions of the adjoint approach have been hampered by their reliance on legacy community codes (e.g., Bunge et al., 2003; Liu

et al., 2010; Shephard et al., 2010; Colli et al., 2018; Ghelichkhan et al., 2021). These codes are not easily extensible to dif-ferent geometries or approximations of the underlying physics, employ solver strategies that have since been superseded, and often have limits on model resolution due to the fully structured discretisations encoded. Moreover, the complex and extensive low-level code implementation of coupled adjoint and forward calculations remains obscured, making it difficult to extend and validate, thus restricting the types of observational datasets that can be incorporated. An example of these datasets are the (un-

certain) constraints from plate tectonic reconstruction models that are prescribed kinematically (e.g., Spasojevic et al., 2009; Shephard et al., 2010; Zhou and Liu, 2017; Colli et al., 2018; Ghelichkhan et al., 2021), as opposed to being formally incor-porated through the misfit functional. The kinematic prescription of the surface velocities, however, can only improve mantle reconstructions forward in time, and therefore prohibits their influence on previous system states. In light of these limitations, there is a need for a general framework that can robustly handle the rheological complexities of Earth's mantle and is easily

extensible and transferable to other problems in mantle and lithosphere dynamics. Such a framework must also be capable of





utilising a variety of observational constraints to comprehensively unravel the historical evolution of Earth's interior's and its diverse impacts at Earth's surface surface (Davies et al., 2023).

In this paper, we introduce the G̲eoscientific A̲dj̲oint O̲ptimisation Pl̲atform (G-ADOPT), research software infrastructure that allows us to overcome these limitations. G-ADOPT is built around three state-of-the-art software libraries — (i) Firedrake, an automated system for solving a range of partial differential equations using the finite element method (Rathgeber et al., 2016), recently validated for geodynamics (Davies et al., 2022); (ii) Dolfin-Adjoint, which automatically derives the discrete adjoint equations in a form compatible with Firedrake (Farrell et al., 2013a; Mitusch et al., 2019); and (iii) the Rapid Optimisation Library (ROL), a Trillions package for performing highly-efficient large-scale optimisation (The ROL Project Team, 2022). When combined, they provide a geodynamic inversion framework that is highly efficient, with forward and adjoint calculations that achieve theoretical computational efficiency.

We structure our paper as follows: in Section 2.1 we describe the geodynamic forward problem and our solution strategy using G-ADOPT. Section 2.2 describes the inverse problem considered herein, where we focus on finding the (unknown) initial condition using an objective functional that accounts for observations of surface velocity over time and the final state temperature field. This is broken into three subsections: (i) Section 2.2.1 describes discrete and continuous approaches for obtaining the adjoint systems followed by a detailed derivation of the discrete adjoint systems; (ii) Section 2.2.2 introduces Dolfin-Adjoint, the underlying approach utilised in G-ADOPT to compute the discrete adjoint and derivative fields; and (iii) Section 2.2.6 provides an overview of the gradient-based optimisation approach utilised here, facilitated by ROL. We demonstrate the applicability of our approach in Section 3, using two sets of twin experiments with increasing complexity, where we reconstruct the spatial and temporal evolution of a reference simulation. The first set of experiments (Section 3.1) involves a simple isoviscous simulation within an enclosed square domain, while the second set (Section 3.2) examines convection with a non-linear (temperature, depth and strain-rate dependent) rheology, at Earth-like convective vigour, within an annular domain, which has direct applicability to convection within Earth's mantle. We finalise by discussing our results and conclusions in Sections 4 and 5, respectively.

## 2 Method

### 2.1 Forward Problem

#### 2.1.1 Governing Equations and Boundary Conditions

Mantle flow is described by conservation laws for mass, momentum and energy. We solve these equations in their simplest form, assuming incompressibility and the Boussinesq approximation. The three non-dimensional conservation equations are

$$-\nabla \cdot (2\eta\dot{\boldsymbol{\epsilon}}(\boldsymbol{u})) + \nabla p + \mathrm{Ra}T\hat{\boldsymbol{k}} = 0, \tag{1a}$$

$$\nabla \cdot \boldsymbol{u} = 0, \tag{1b}$$

$$\frac{\partial T}{\partial t} + \boldsymbol{u} \cdot \nabla T - \kappa\nabla^2 T - H = 0, \tag{1c}$$



| Symbol | Description | Symbol | Description |
|--------|-------------|--------|-------------|
| $\eta$ | Dynamic viscosity | $\rho_0$ | Reference density |
| $\hat{\boldsymbol{k}}$ | Unit vector in upward gravity direction | $\alpha$ | Thermal expansion coefficient |
| $H$ | Internal heating rate | $\Delta T$ | Characteristic temperature change |
| $\dot{\epsilon}$ | Strain-rate | $g$ | Gravitational acceleration |
| Ra | Rayleigh number | $d$ | Characteristic length |
| $\kappa$ | Thermal diffusivity | $\mu_0$ | Reference dynamic viscosity |

**Table 1.** Symbols used in this study.

with the vector field $\boldsymbol{u}$, and scalar fields $p$ and $T$ as the principal unknowns of velocity, pressure and temperature, respectively. Table 1 summarises other symbols used in Eqs. 1 and elsewhere. In Eq. 1a, the strain-rate tensor $\dot{\boldsymbol{\epsilon}}(\boldsymbol{u})$ is given by

$$\dot{\boldsymbol{\epsilon}}(\boldsymbol{u}) = \frac{1}{2}\left(\nabla\boldsymbol{u} + (\nabla\boldsymbol{u})^T\right), \tag{2}$$

and the Rayleigh number is defined by

$$\mathrm{Ra} = \frac{\rho_0 \alpha \Delta T g d^3}{\mu_0 \kappa}. \tag{3}$$

For this problem, we define the time interval of interest as $I = [t_I, t_F]$, with the computational domain $V$ bounded by $\partial V$, and $S$ and $C$ denoting the top and bottom boundaries, respectively. For all simulations, free-slip and impermeable velocity boundary conditions are specified on all boundaries, whilst temperature boundary conditions are set to constant values of $T_S$ and $T_C$, at top and bottom boundaries, respectively. For the simulations considered in an enclosed square domain, natural temperature boundary conditions (zero heat-flux) are specified on side walls. The set of boundary conditions, for top and bottom boundaries, are listed in Eqs. 4:

$$\boldsymbol{u}(x,t) \cdot \boldsymbol{n} = 0, \qquad\qquad\qquad x \in \partial V,\ t \in I, \tag{4a}$$

$$[\eta\,\dot{\boldsymbol{\epsilon}}\,(\boldsymbol{u}\,(x,t)) \cdot \boldsymbol{n}] \cdot \boldsymbol{s} = 0, \qquad\qquad x \in \partial V,\ t \in I, \tag{4b}$$

$$T(x,t) = T_S, \qquad\qquad\qquad x \in S,\ t \in I, \tag{4c}$$

$$T(x,t) = T_C, \qquad\qquad\qquad x \in C,\ t \in I. \tag{4d}$$

In Eqs. 4, $\boldsymbol{n}$ denotes the outer normal vector, and $\boldsymbol{s}$ any tangential vector. Finally, as mantle convection is an initial value problem we require a prescribed temperature field at initial time $t_I$:

$$T(x,t_I) = T_{IC}(x), \qquad x \in V. \tag{5}$$

### 2.1.2 Solution Strategy: Leveraging Firedrake through G-ADOPT

We use the Finite Element method to solve the coupled system of partial differential equations presented in Eqs. 1. For the Stokes system, we use a $Q_2 - Q_1$ (piecewise biquadratic, resp. bilinear) finite element pair for velocity and pressure, with



$Q_2$ elements used for temperature. We strongly impose Dirichlet boundary conditions for temperature at top and bottom
boundaries. Free-slip velocity boundary conditions are imposed in two ways: (i) in our square domain cases, we impose
strong Dirichlet boundary conditions for $\boldsymbol{u}$; (ii) in our annular domain cases, where boundaries do not align with Cartesian
directions, we employ a symmetric Nitsche penalty method (Nitsche, 1971), which weakly enforces boundary conditions via a
modification of the variational formulation. An implicit mid-point scheme is used for time integration in the energy equation.

To solve the coupled system of forward (and adjoint) equations, we employ Firedrake, which provides an automated system
designed for the solution of partial differential equations using the Finite Element method (Ham et al., 2023). It incorporates
principles and some aspects of the code-base from the FEniCS project (Logg et al., 2012), including its use of the Unified Form
Language (UFL) (Alnæs et al., 2014) for the representation of variational problems. UFL is a high-level language utilised
for the symbolic description of the governing equations in a form that closely mimics ther mathematical formulation. The
key advantage of UFL in this work lies in its high-level abstraction, which allows for an innovative automatic derivation
approach when deriving adjoint systems. We will address the significance of UFL in automatic derivation of adjoint systems
in Section 2.2.2.

Firedrake provides an array of features that are particularly conducive to tackling problems in geophysical fluid dynamics.
Key among these features is support for a variety of Finite Element discretisations, including a highly efficient implementation
of discretisations based on extruded meshes, programmable non-linear solvers and operator-aware solver preconditioners that
can be combined in a flexible manner to create linear or non-linear systems, which are solved by PETSc (e.g., Balay et al., 1997;
Dalcin et al., 2011; Balay et al., 2023). The suitability of Firedrake for geodynamics has been demonstrated via comparison
with a comprehensive set of analytical solutions and community benchmarks in Davies et al. (2022). We refer the reader to
this study, alongside Rathgeber et al. (2016), for a more in-depth discussion on Firedrake and its dependencies, alongside an
outline of the solution strategy employed herein.

## 2.2 Inverse problem: An Optimisation Approach

Representative reconstructions of the spatial and temporal evolution of mantle flow require knowledge of the initial condition
(i.e. the thermochemical state of the mantle at some point in the past), the determination of which is an inverse problem. We
therefore seek the best fitting initial condition ($T_{IC}$), that results in the minimum of an objective functional that measures
the difference between predictions and observations of mantle states and irregularity in the solutions. We use the following



mathematical description of the objective functional:

$$J = \frac{1}{2} \int_I \int_V (T - T_{obs})^2 \, \delta(t - t_F) \, \mathrm{d}x \, \mathrm{d}t$$

$$+ \frac{\beta_u}{2} \int_I \int_S (\boldsymbol{u} - \boldsymbol{u}_{obs})^2 \, \mathrm{d}s \, \mathrm{d}t$$

$$+ \frac{\beta_d}{2} \int_I \int_V G(r) \left(T - \bar{T}\right)^2 \delta(t - t_I) \, \mathrm{d}x \, \mathrm{d}t$$

$$+ \frac{\beta_s}{2} \int_I \int_V \left[\nabla \left(T - \bar{T}\right)\right]^2 \delta(t - t_I) \, \mathrm{d}x \, \mathrm{d}t \tag{6}$$

The first term in Eq. 6 accounts for the misfit between model predictions and temperature recorded at the final instance, whilst the second term measures the misfit in surface velocities through time. The other terms are Tikhonov regularisation components (Tikhonov, 1963; Hansen, 1992), the first of which penalises deviations from an a-priori depth-averaged profile, $\bar{T}$ (i.e. the damping term), and the second penalises the gradient of these deviations to produce less complex solutions (i.e. the smoothing term). For the damping term, we employ a depth-dependent pre-factor, $G(r)$, that is zero within the mid mantle but

transitions to one in the thermal boundary layers. In these areas, lateral material transport and diffusive processes dominate, leading to diminished sensitivity between the choice of initial condition and the final temperature field. $G(r)$, therefore, helps to minimise the amplitude of the solution within the top and bottom thermal boundary layers, removing noise that would otherwise diffuse over the simulation. The three weighting terms, $\beta_u$, $\beta_s$ and $\beta_d$ in Eq. 6 are defined as,

$$\beta_u = \alpha_u \frac{\int_V T_{obs}^2 \, \mathrm{d}x}{\Delta t \int_S \left(u_{obs}^{t=t_F}\right)^2 \mathrm{d}x}, \tag{7a}$$

$$\beta_d = \alpha_d \frac{\int_V T_{obs}^2 \, \mathrm{d}x}{\int_V \left(\bar{T}\right)^2 \mathrm{d}x}, \tag{7b}$$

$$\beta_s = \alpha_s \frac{\int_V T_{obs}^2 \, \mathrm{d}x}{\int_V \left(\nabla \bar{T}\right)^2 \mathrm{d}x}. \tag{7c}$$

Note that $\Delta t$ is the total duration of the simulation. The integrals in Eqs. 7 are employed to ensure normalised objective terms relative to the final temperature misfit (first term on the right hand side of Eq. 6). The three scaling parameters, $\alpha_u$, $\alpha_s$ and $\alpha_d$, can be set to adjust the importance of these terms relative to the final temperature misfit. We perform a parameter search to find

the best performing combinations of $\alpha$ values.

    For our inverse problem we are interested in optimising $J$ described by Eq. 6. $J$ depends on some 'control' parameters, which in this case is the initial temperature field, and some 'state' variables (i.e. surface velocity and the final temperature), which are solutions of the forward problem in Eqs. 1, with the forward system itself depending again on the control. To solve this problem, we define a 'reduced' functional, which is a function of the initial condition, $T_{IC}$, alone. This reduced functional

is typically defined by first solving the forward PDEs (Eqs. 1) for a given value of the control, and then substituting the solutions into the expression for the functional (Eq. 6). The result is a functional that depends only on the control parameters, not the state variables directly (hence the name reduced).





Non-linear optimisation methods provide the means to find the optimal $T_{IC}$ by minimising a reduced functional defined by $J$. These methods are iterative. They begin with an initial guess of $T_{IC}$ to generate a sequence of improved estimates (called *iterates*) until certain conditions, e.g., residual tolerance, in the objective functional are achieved. Crucial for the efficiency of these iterations are derivatives of Eq. 6 with respect to $T_{IC}$. Owing to the large number of unknowns in 3-D spherical mantle convection models with Earth-like parameters, obtaining the derivative by means of classical finite differencing techniques is impractical. The *adjoint* method serves as a mathematically elegant and computationally efficient way to obtain the derivatives (e.g., Giles and Pierce, 2000; Plessix, 2006; Hinze et al., 2008).

### 2.2.1 Continuous versus Discrete Adjoints

Approaches to deriving, implementing, and obtaining derivatives using the adjoint method primarily fall into two categories: (i) continuous, or the differentiate-then-discretise approach; and (ii) discrete, or the discretise-then-differentiate approach (Gunzburger, 2000).

The continuous approach commences with derivation of the adjoint equations. The resemblance between the forward and resulting adjoint equations allows the adjoint PDEs to be discretised in a consistent manner and, subsequently, implemented within a numerical framework. By deriving the continuous adjoint PDEs, one can develop an understanding of the key characteristics of adjoint sensitivities, the physical implications of individual terms in the PDEs, and their boundary conditions. Moreover, the continuous method affords complete autonomy in the discretisation and implementation of the adjoint system, often leading to simplified, but more cost-effective approximations of the solutions, as demonstrated by Ismail-Zadeh et al. (2004).

By contrast, the discrete approach relies on already discretised forward equations then differentiates and transposes them to obtain the adjoint equations. This method's primary advantage lies in maintaining consistency in spatial and temporal discretisations, allowing for the automatic determination of the exact gradient of the (discrete) objective functional (Gunzburger, 2002). Such consistency ensures full convergence of second-order Newtonian optimisation methods and simplifies the debugging of adjoint programs (Giles and Pierce, 2000). For example, with the discrete approach, even minor inconsistencies in the derivative can highlight numerical or programming errors that must be rectified. It also permits the automatic creation of the adjoint program, stemming from the property that a transposed (adjoint) matrix shares the same eigenvalues with the original linear matrix, ensuring convergence for the adjoint problem's iterative solution methods (Giles and Pierce, 2000). This advantage has facilitated the development of various Automatic Differentiation (AD) tools in recent decades, including those used in Tensor-Flow (Abadi et al., 2015) and PyTorch (Paszke et al., 2019). It is essential to note that the continuous and discrete methods are equivalent in the limit of infinite spatial and temporal resolution. However, in practical terms, it is the discrete method that typically provides more accurate gradient information. For more details on both approaches we refer the reader to Giles and Pierce (2000) and Gunzburger (2002).





### 2.2.2 Dolfin-Adjoint

Robust and efficient derivative calculations for large-scale simulations using automatic differentiation is challenging and often too slow for the purpose of large-scale optimisation problems (Naumann, 2011). This inefficiency is often attributed to the usually employed approach of treating a numerical model as a sequence of elementary instructions such as addition, multiplication or exponentiation, known as *blocks*. Once the AD tool establishes a sequence of blocks with their dependencies (a process often called *taping*), each block is individually differentiated, and one arrives at the derivative of the entire model using the
chain rule.

Dolfin-Adjoint uses an innovative approach to achieve theoretical efficiency by using so-called operator overloading differentiation (Tijskens et al., 2002). By leveraging the high-level mathematical language used by Firedrake and FEniCS (UFL), Dolfin-Adjoint performs the taping process at the highest abstraction level. This can result in blocks that symbolise whole PDE system solves, for which the adjoint derivation is performed at the same level of abstraction. The derived adjoint operation
to a block is itself a Firedrake operation. This facilitates generation of the low-level adjoint code being generated using the same finite element form compiler as the forward model. That is, while the taping operation is in essence similar to the fundamental abstraction in automatic differentiation techniques, Dolfin-Adjoint operates at a much higher level of abstraction and, accordingly, can achieve maximum efficiency and robustness.

### 2.2.3 Discrete forward model

As explained in more detail in Davies et al. (2022), the forward geodynamical model can be described as a series of linear and nonlinear solves. Although we solve for temperature using $Q_2$ elements, we choose the control $T_{IC}$ to be in the $Q_1$ function space as a means to regularise the inversion problem. This means we need to project $T_{IC}$ to the discrete function space $Q = Q_2$, to obtain a temperature $T^0$ used in the first timestep. This can be formulated as solving the following system for $T^0$:

$$F_{\text{project}}(q; T^0, T_{IC}) := \int_V q\left(T^0 - T_{IC}\right) \mathrm{d}x = 0 \quad \text{for all } q \in Q, \tag{8}$$

where $q$ are test functions in $Q$. Subsequently we solve in each timestep $n = 0, \dots N-1$, the following two systems for $\boldsymbol{u}^n, p^n$, and $T^{n+1}$, respectively:

$$F_{\text{Stokes}}(\boldsymbol{v}, w; \boldsymbol{u}^n, p^n, T^n) := \int_V (\nabla \boldsymbol{v}) : \eta(\boldsymbol{u}^n, T^n)\left[\nabla \boldsymbol{u}^n + (\nabla \boldsymbol{u}^n)^T\right] \mathrm{d}x - \int_V (\nabla \cdot \boldsymbol{v}) p^n \mathrm{d}x - \int_V \text{Ra} T^n \boldsymbol{v} \cdot \hat{\boldsymbol{k}} \mathrm{d}x$$
$$- \int_V w \nabla \cdot \boldsymbol{u}^n \mathrm{d}x = 0 \quad \text{for all } \boldsymbol{v} \in V, w \in W, \tag{9}$$

$$F_{\text{energy}}(q; T^{n+1}, T^n, \boldsymbol{u}^n) := \int_V q\frac{T^{n+1} - T^n}{\Delta t}\mathrm{d}x + \int_V q\boldsymbol{u}^n \cdot \nabla T^{n+\theta}\mathrm{d}x + \int_V (\nabla q) \cdot \left(\kappa \nabla T^{n+\theta}\right)\mathrm{d}x = 0 \quad \text{for all } q \in Q, \tag{10}$$

where $V, W$ are the discrete function spaces for velocity and pressure (here $V = [Q_2]^{\text{dim}}, W = Q_1$) with test functions $\boldsymbol{v}$, and
270 $w$. $T^{n+\theta}$ is the weighted average $\theta T^{n+1} + (1-\theta)T^n$. Note that we assume a strain rate and temperature dependent rheology,





and thus write $\eta = \eta(\boldsymbol{u}, T)$. This makes Eq. (9) a non-linear system, which we solve through Newton's method. The discrete functional is calculated as

$$J(T^N, \boldsymbol{u}^0, \ldots, \boldsymbol{u}^{N-1}, T_{IC}) = \frac{1}{2} \int\limits_V \left(T^N - T_{obs}\right)^2 \mathrm{d}x + \frac{\beta_u}{2} \sum_{n=0}^{N-1} \int\limits_S \left(\boldsymbol{u}^n - \boldsymbol{u}^n_{obs}\right)^2 \mathrm{d}s$$
$$+ \frac{\beta_d}{2} \int\limits_V G(r) \left(T_{IC} - \bar{T}\right)^2 \mathrm{d}x + \frac{\beta_s}{2} \int\limits_V \left[\nabla(T_{IC} - \bar{T})\right]^2 \mathrm{d}x \tag{11}$$

### 2.2.4 Calculating Gradients Using the Adjoint Method

We denote the entire forward solution as $z = (T^0, \ldots T^N, \boldsymbol{u}^0, \ldots \boldsymbol{u}^{N-1}, p^0, \ldots p^{N-1})$, so that the functional can be written as a function $J(z, T_{IC})$, of $z$ and the control $T_{IC}$. The forward solution itself is also dependent on $T_{IC}$, as for each choice of $T_{IC}$ we can solve the forward model to obtain $z(T_{IC})$. We define the reduced functional $\hat{J}$ as

$$\hat{J}(T_{IC}) = J(z(T_{IC}), T_{IC}). \tag{12}$$

Thus we can reformulate the PDE-constrained minimisation problem:

*minimise $J(z, T_{IC})$ under the constraints (8–10),*

as an unconstrained minimisation problem for $\hat{J}(T_{IC})$. To use efficient gradient-based optimisation algorithms we do, however, need a means of computing its gradient for which we will employ the adjoint method. In addition to the forward solution $z$, we also define an adjoint solution $\lambda = (\Psi^0, \ldots \Psi^N, \phi^0, \ldots \phi^{N-1}, \xi^0 \ldots \xi^{N-1})$, where each component is associated with one of the constraints: $\Psi^0 \in Q$ with (8), $\phi^n \in V, \xi^n \in W$ with (9), and $\Psi^{n+1} \in Q$ with (10) for $n = 0, \ldots N-1$. Using these we define the following sum of the constraints, with each constraint weighted by the corresponding adjoint solution $\lambda$:

$$F(\lambda; z, T_{IC}) = F_{\text{project}}(\Psi^0; T^0, T_{IC}) + \sum_{n=0}^{N-1} F_{\text{Stokes}}(\phi^n, \xi^n; \boldsymbol{u}^n, p^n, T^n) + \sum_{n=0}^{N-1} F_{\text{energy}}(\Psi^{n+1}; T^{n+1}, T^n, \boldsymbol{u}^n). \tag{13}$$

Since, by definition, for any choice of $T_{IC}$ the associated forward solution $z(T_{IC})$ satisfies all constraints, we have:

$$F(\lambda; z(T_{IC}), T_{IC}) = 0,$$

and thus, for any choice of $\lambda$ we have:

$$\frac{\partial F(\lambda; z(T_{IC}), T_{IC})}{\partial T_{IC}} = \left.\frac{\partial F(\lambda; z, T_{IC})}{\partial z}\right|_{z=z(T_{IC})} \frac{\partial z(T_{IC})}{\partial T_{IC}} + \left.\frac{\partial F(\lambda; z, T_{IC})}{\partial T_{IC}}\right|_{z=z(T_{IC})} = 0. \tag{14}$$

If we choose $\lambda$ to be the solution to the following, so called, *adjoint equation*

$$\left.\frac{\partial F(\lambda; z, T_{IC})}{\partial z}\right|_{z=z(T_{IC})} = \left.\frac{\partial J(z, T_{IC})}{\partial z}\right|_{z=z(T_{IC})}, \tag{15}$$



we can work out the gradient of the reduced functional:

$$
\frac{\partial \hat{J}(T_{IC})}{\partial T_{IC}} = \left.\frac{\partial J(z, T_{IC})}{\partial z}\right|_{z=z(T_{IC})} \frac{\partial z(T_{IC})}{\partial T_{IC}} + \left.\frac{\partial J(z, T_{IC})}{\partial T_{IC}}\right|_{z=T_{IC}} \tag{16}
$$

$$
= \left.\frac{\partial F(\lambda; z, T_{IC})}{\partial z}\right|_{z=z(T_{IC})} \frac{\partial z(T_{IC})}{\partial T_{IC}} + \left.\frac{\partial J(z, T_{IC})}{\partial T_{IC}}\right|_{z=T_{IC}} \tag{17}
$$

$$
= -\left.\frac{\partial F(\lambda; z, T_{IC})}{\partial T_{IC}}\right|_{z=z(T_{IC})} + \left.\frac{\partial J(z, T_{IC})}{\partial T_{IC}}\right|_{z=T_{IC}} \tag{18}
$$

### 2.2.5 Discrete Backward Model

Although the adjoint equation (15) is derived symbolically from the forward model (8–10), and solved for fully automatically by Dolfin-Adjoint, we here briefly work out the discrete adjoint equations to show that these equations can still be interpreted as the solution process of a backwards-in-time PDE, similar to the continuous adjoint approach, but that a specific time-discretisation is derived, which is necessary to obtain a gradient that is consistent with the discrete forward model. Split out by component, the adjoint equations read

$$
\frac{\partial F}{\partial T^0} = \frac{\partial J}{\partial T^0}, \tag{19}
$$

$$
\frac{\partial F}{\partial \boldsymbol{u}^n} = \frac{\partial J}{\partial \boldsymbol{u}^n}, \quad \text{for } n = 0, \ldots N-1, \tag{20}
$$

$$
\frac{\partial F}{\partial p^n} = \frac{\partial J}{\partial p^n}, \quad \text{for } n = 0, \ldots N-1, \tag{21}
$$

$$
\frac{\partial F}{\partial T^{n+1}} = \frac{\partial J}{\partial T^{n+1}}, \quad \text{for } n = 0, \ldots N-1, \tag{22}
$$

which can be solved for $\Psi^n, \phi^n, \xi^n$.

These equations can be solved by going backwards through the timesteps $n = N-1 \ldots 0$. In each, we first solve (22) for $\Psi^{n+1}$. Starting at the last timestep $n = N-1$, we get:

$$
\frac{\partial F_{\text{energy}}(\Psi^N; T^N, T^{N-1}, \boldsymbol{u}^{N-1})}{\partial T^N} \delta T = \frac{\partial J}{\partial T^N} \delta T, \tag{23}
$$

where we have applied the gradient of $F$ with respect to $T^N$ to an arbitrary perturbation $\delta T$. This allows us to interpret this equation as a weak form, tested with $\delta T$, that we can solve for $\Psi^N$:

$$
\int_V \Psi^N \frac{\delta T}{\Delta t} \mathrm{d}x + \theta \int_V \Psi^N \boldsymbol{u}^{N-1} \cdot \nabla \delta T \mathrm{d}x + \theta \int_V \left(\nabla \Psi^N\right) \cdot \left(\kappa \nabla \delta T\right) \mathrm{d}x = \int_V \left(T^N - T_{obs}\right) \delta T \mathrm{d}x \quad \text{for all } \delta T \in Q. \tag{24}
$$

Defining an auxiliary $\Psi^{N+1} = \Delta t(T^N - T_{obs})$, and integrating the advection term by parts, we get:

$$
\int_V \left[\frac{\Psi^N - \Psi^{N+1}}{\Delta t} \delta T - \theta \nabla \cdot \left(\Psi^N \boldsymbol{u}^{N-1}\right) \delta T + \theta \left(\nabla \Psi^N\right) \cdot \kappa \nabla \delta T\right] \mathrm{d}x = 0 \quad \text{for all } \delta T \in Q, \tag{25}
$$

which for $\theta = 1$, we may recognise an advection-diffusion equation run backwards in time.



For $n < N - 1$, equation (22) contains more terms:

$$\frac{\partial F_{\text{energy}}(\Psi^{n+1}; T^{n+1}, T^n, \boldsymbol{u}^n)}{\partial T^{n+1}} \delta T = -\frac{\partial F_{\text{energy}}(\Psi^{n+2}; T^{n+2}, T^{n+1}, \boldsymbol{u}^{n+1})}{\partial T^{n+1}} \delta T$$

$$-\frac{\partial F_{\text{Stokes}}(\boldsymbol{\phi}^{n+1}, \xi^{n+1}; \boldsymbol{u}^{n+1}, p^{n+1}, T^{n+1})}{\partial T^{n+1}} + \frac{\partial J}{\partial T^{n+1}} \delta T \quad \text{for all } \delta T \in Q. \quad (26)$$

as the energy equation in both forward time steps $n$ and $n+1$ depends on $T^{n+1}$, as does the Stokes system in time step $n+1$. Going backwards through the equations however, we can still solve for $\Psi^{n+1}$, associated with time step $n$, as we have already solved for $\boldsymbol{\phi}^{n+1}, \xi^{n+1}$, and $\Psi^{n+2}$ associated with time step $n+1$. Note that the $\partial J/\partial T^{n+1}$–term vanishes in this case as $J$ does not explicitly depend on intermediate temperature solutions. Similar to (25), we may rewrite to

$$\int_V \left[ \frac{\Psi^{n+1} - \Psi^{n+2}}{\Delta t} \delta T - \nabla \cdot \left( \theta \Psi^{n+1} \boldsymbol{u}^n + (1-\theta) \Psi^{n+2} \boldsymbol{u}^{n+1} \right) \delta T + \nabla \left( \theta \Psi^{n+1} + (1-\theta) \Psi^{n+2} \right) \cdot \kappa \nabla \delta T \right] \mathrm{d}x =$$

$$-\int_V \nabla \boldsymbol{\phi}^{n+1} : \frac{\partial \eta(\boldsymbol{u}^{n+1}, T^{n+1})}{\partial T^{n+1}} \left[ \nabla \boldsymbol{u}^{n+1} + \left( \nabla \boldsymbol{u}^{n+1} \right)^T \right] \delta T \mathrm{d}x + \int_V \text{Ra} \boldsymbol{\phi}^{n+1} \cdot \hat{\boldsymbol{k}} \delta T \mathrm{d}x \quad \text{for all } \delta T \in Q, \quad (27)$$

which we can interpret as a backward-in-time theta-weighted advection diffusion step for $\Psi$, with source terms associated with sensitivity of the rheology and buoyancy to temperature. Note, however, that here in the advection term we weight both $\Psi$ and $\boldsymbol{u}$ with $\theta$, in contrast to the forward timestep (10).

After solving for $\Psi^{n+1}$, we can solve (20) together with (21) for $\boldsymbol{\phi}^n$ and $\xi^n$:

$$\frac{\partial F_{\text{Stokes}}(\boldsymbol{\phi}^n, \xi^n; \boldsymbol{u}^n, p^n, T^n)}{\partial \boldsymbol{u}^n} \delta \boldsymbol{u} + \frac{\partial F_{\text{Stokes}}(\boldsymbol{\phi}^n, \xi^n; \boldsymbol{u}^n, p^n, T^n)}{\partial p^n} \delta p = -\frac{\partial F_{\text{energy}}(\Psi^{n+1}; T^{n+1}, T^n, \boldsymbol{u}^n)}{\partial \boldsymbol{u}^n} \delta \boldsymbol{u} + \frac{\partial J}{\partial \boldsymbol{u}^n} \delta \boldsymbol{u} \quad (28)$$

for all velocity and pressure perturbations $\delta \boldsymbol{u} \in V, \delta p \in W$. This leads to the following weak-form equation for $\boldsymbol{\phi}^n$ and $\xi^n$:

$$\int_V \left[ \nabla \boldsymbol{\phi}^n + (\nabla \boldsymbol{\phi}^n) \right] : \eta(\boldsymbol{u}^n, T^n) \nabla \delta \boldsymbol{u} \mathrm{d}x + \int_V \left[ \nabla \boldsymbol{\phi}^n + (\nabla \boldsymbol{\phi}^n) \right] : \frac{\partial \eta(\boldsymbol{u}^n, T^n)}{\partial \boldsymbol{u}^n} \delta \boldsymbol{u} \nabla \boldsymbol{u} \mathrm{d}x - \int_V \xi^n \nabla \cdot \delta \boldsymbol{u}^n \mathrm{d}x - \int_V (\nabla \cdot \boldsymbol{\phi}^n) \delta p \mathrm{d}x$$

$$= -\int_V \Psi^{n+1} \delta \boldsymbol{u} \cdot \nabla T^{n+\theta} + \beta_u \int_S (\boldsymbol{u}^n - \boldsymbol{u}_{obs}) \delta \boldsymbol{u} \mathrm{d}x \quad \text{for all } \delta \boldsymbol{u} \in V, \delta p \in W \quad (29)$$

The left-hand side is similar to the Stokes system in (9), except for an additional $\partial \eta/\partial \boldsymbol{u}$ term, alongside the fact that this is now just a linear system as the rheology only depends on the forward variables $\boldsymbol{u}^n$ and $T^n$. Instead of a buoyancy term, we now have forcing terms associated with sensitivity of temperature advection and the mismatch with observed velocities at the surface.

Finally, after having solved either (25) (n=N-1) or (27) (n<N-1) for $\Psi^{n+1}$, and (29) for $\boldsymbol{\phi}^n, \xi^n$ going backward through the timesteps $n = N - 1 \rightarrow n = 0$, we can solve (19) for $\Psi^0$:

$$\frac{\partial F_{\text{project}}(\Psi^0; T^0, T_{IC})}{\partial T^0} \delta T = -\frac{\partial F_{\text{energy}}(\Psi^1, T^1, T^0, \boldsymbol{u}^0)}{\partial T^0} \delta T - \frac{\partial F_{\text{Stokes}}(\boldsymbol{\phi}^0, \xi^0; \boldsymbol{u}^0, p^0, T^0)}{\partial T^0} \delta T \quad \text{for all } \delta T \in Q \quad (30)$$





which can be worked out as a projection

$$\int_V \Psi^0 \delta T \mathrm{d}x = \int_V \frac{\Psi^1}{\Delta t} \delta T \mathrm{d}x + (1-\theta) \int_V \left(\nabla \cdot \Psi^1 \boldsymbol{u}^1\right) \delta T \mathrm{d}x - (1-\theta) \int_V \nabla \psi^{n+1} \cdot \kappa \nabla \delta T \mathrm{d}x$$
$$- \int_V \nabla \boldsymbol{\phi}^1 : \frac{\partial \eta(\boldsymbol{u}^1, T^1)}{\partial T^1} \left[\nabla \boldsymbol{u}^1 + \left(\nabla \boldsymbol{u}^1\right)^T\right] \delta T \mathrm{d}x + \int_V \mathrm{Ra} \boldsymbol{\phi}^1 \cdot \hat{\boldsymbol{k}} \delta T \mathrm{d}x \quad \text{for all } \delta T \in Q. \quad (31)$$

Finally, the gradient of the reduced functional with respect to the control is obtained by:

$$\frac{\partial \hat{J}(T_{IC})}{\partial T_{IC}} \delta T = - \int_V \Psi^0 \, \delta T \, \mathrm{d}x + \frac{\beta_d}{2} \int_V \left(T_{IC} - \bar{T}\right) \delta T \, \mathrm{d}x + \frac{\beta_s}{2} \int_V \nabla \left(T_{IC} - \bar{T}\right) \cdot \nabla \delta T \, \mathrm{d}x \quad (32)$$

### 2.2.6 Gradient-Based Non-linear Optimisation

To find the solution to the inverse problem, the gradient fields from Firedrake and Dolfin-Adjoint can be redirected to an optimisation package with a Python interface (e.g., scipy.optimize: Virtanen et al., 2020). However, the majority of well-established optimisation packages are hard-coded to apply the Euclidean ($l^2$) inner product for optimisation-specific operations. $l^2$, typi-
cally referred to as a sequence space, is often used in signal processing or discrete mathematics where functions are treated as sequences, i.e. discrete data, and do not represent continuous functions. The Euclidean inner product is therefore not suitable for finite element function-based optimisation and, unlike $L^2$ or Sobolev spaces, it cannot produce mesh and/or basis function independent convergence (Schwedes et al., 2017). The Rapid Optimisation Library (ROL, The ROL Project Team, 2022), a Trilinos package for large-scale optimisation, resolves this issue by introducing a generic interface for data structures that
can be overloaded to perform inner-product aware operations (e.g., in $L^2$ space) and achieve mesh-independent convergence results (Schwedes et al., 2016). ROL has been primarily used for the solution of optimal design, optimal control and inverse problems in large-scale engineering applications (Iglesias et al., 2018; Kouri et al., 2021a, b, 2023) and has a comprehensive catalogue of gradient-based optimisation algorithms. For the results presented herein, we use the python interface for ROL in Firedrake, which we have supplemented with additional checkpointing functionality. This allows us to checkpoint intermediate
optimisation states and variables, including those related to step lengths or Hessian estimates, and subsequently restart the optimisation procedure without loss of performance. This, in particular, is relevant on modern High Performance Computing facilities with strict wall time limits.

The distinguishing factor between different optimisation algorithms (in ROL or elsewhere) is the strategy used to move from one iterate to the next. Broadly speaking, there are two strategies for moving from the current iterate, $x_k$, to a new one,
$x_{k+1}$: line-search and trust-region strategies (Nocedal and Wright, 1999). A fundamental distinction between line-search and trust-region methods lies in the sequence of selecting the direction and distance (Nocedal and Wright, 2006). In line search methods, the direction is initially fixed, and an appropriate step length is subsequently determined. Conversely, trust-region methods commence by establishing an initial size for a trusted region (hence the name), and then simultaneously constrain the direction and step to achieve sufficient amount of improvement within this trusted region. The size of this trusted region
around the current iterate is determined according to a model, $m$, that approximates the region around the current iterate, $x_k$,





according to a quadratic approximation. At each iteration, the accuracy of this model then is assessed based on its agreement with the actual changes in the function, $f$. If the new value, $f(x_k + p_k)$ is greater than the current value of $f(x_k)$, $m$ is not a good approximation of the objective functional around $x_k$ and the size of the trusted-region measured by the trust-region *radius* is therefore reduced to improve the applicability of the model. By bounding the calculations to a trusted region where

model $m$ is applicable, trust-region methods prohibit overly aggressive steps, which make them suitable for handling negative curvature situations (non-convexity optimisation problems) more gracefully compared to line-search methods. Nonetheless, faster convergence can be achieved by expanding trust region in case of a predictive model, ensuring robust minimisation of the objective functional.

     In this study, we employ the trust region method of Lin and Moré (1999) implemented in ROL. Lin-More is a truncated New-

ton method, consisting of repeated application of an iterative algorithm to approximately solve Newton's equations (Dembo and Steihaug, 1983). Lin-More can effectively handle provided bound constraints by ensuring that variables remain within their specified bounds: At each iteration, variables are classified into "active" and "inactive" sets. Variables at their bounds and not allowing descent are considered active and are fixed during the iteration. The remaining variables, which can change without violating the bounds, are inactive. The described properties renders the algorithm as a robust and efficient method for solving

bound-constrained optimisation problems.

## 3   Numerical Experiments and Results

Twin experiments serve as a means to illustrate the feasibility of geophysical inverse methods. In our experimental setup, we generate a synthetic reference simulation that advances forward in time, starting from a user-defined initial condition. We use this reference simulation to emulate a real-Earth reconstruction scenario, where the resulting temperature field at the

final time, $T(x, t = t_F)$, and the corresponding surface velocities at all times, $u(x = x_S, t)$, are stored for subsequent use as 'observations' in reconstructing the initial state of the mantle and its evolution through space and time. These fields are used to mimic fundamental datasets in mantle reconstruction models, drawing parallels to 3-D models of mantle temperature inferred from seismic tomography images, and surface velocities derived from plate tectonic reconstructions. We examine two sets of 2-D twin experiments: (i) simulations of a single upwelling in an enclosed square domain, with an isoviscous rheology; and

(ii) convection within an annular domain, incorporating a non-linear visco-plastic rheology, as has previously been employed to generate plate-like behaviour in mantle convection models (see Coltice et al., 2017, for a review).

### 3.1   A Single Upwelling in an Enclosed Square Domain

#### 3.1.1   Forward Problem

We start our experiments by reconstructing the evolution of a single upwelling plume within an enclosed square computational

domain (free-slip boundary conditions on all boundaries). We assume an isoviscous rheology, incompressible flow under the Bousinessq approximation assumption, and a Rayleigh number of $Ra = 10^6$. The model is heated from below ($T_C = 1$) and




```
1: from gadopt import *
2:
3: # load mesh
4: mesh1d = IntervalMesh(150, length_or_left=0.0, right=1.0)  # Unit interval mesh
5: mesh_temp = ExtrudedMesh(
6:           mesh1d, layers=150, layer_height=1.0/150, extrusion_type="uniform")
7:
8: # Set up function spaces for the Q2Q1 pair for velocity+pressure and Q2 for temperature
9: V = VectorFunctionSpace(mesh, "CG", 2)  # Velocity function space (vector)
10: W = FunctionSpace(mesh, "CG", 1)  # Pressure function space (scalar)
11: Z = MixedFunctionSpace([V, W])  # mixed space for Stokes
12: Q = FunctionSpace(mesh, "CG", 2)  # Temperature function space (scalar)
13:
14: z = Function(Z)  # A field over the mixed function space Z
15: u, p = z.subfunctions  # Symbolic UFL expressions for u and p
16:
17: T = Function(Q, name="Temperature")  # Temperature Field
18: X = SpatialCoordinate(mesh)  # Spatial Coordinates
19: # Initial condition
20: T.interpolate(
21:     0.5 * (erf((1 - X[1]) * 3.0) + erf(-X[1] * 3.0) + 1)
22:     + 0.1 * exp(-0.5 * ((X - as_vector((0.5, 0.2))) / Constant(0.1)) ** 2))
23:
24: Z_nullspace = create_stokes_nullspace(Z, closed=True, rotational=False)  # Pressure nullspace
25:
26: checkpoint_file = CheckpointFile("Checkpoint_State.h5", "w")
27: checkpoint_file.save_mesh(mesh)  # saving the mesh
28: Ra = Constant(1e6)  # Rayleigh number
29: approximation = BoussinesqApproximation(Ra)  # Incompressible (Boussinesq) approximation
30: delta_t = Constant(4e-6)  # Constant time step
31: max_timesteps = 80  # Number of time steps
32:
33: # Boundary conditions
34: stokes_bcs = {"bottom": {"uy": 0}, "top": {"uy": 0}, 1: {"ux": 0}, 2: {"ux": 0}}  # Velocity BCs
35: temp_bcs = {"bottom": {"T": 1.0}, "top": {"T": 0.0}}  # Temperature BCs
36:
37: energy_solver = EnergySolver(T, u, approximation, delta_t, ImplicitMidpoint, bcs=temp_bcs)
38: stokes_solver = StokesSolver(z, T, approximation, bcs=stokes_bcs,
39:                     nullspace=Z_nullspace, transpose_nullspace=Z_nullspace)
40:
41: # time loop:
42: for timestep in range(0, max_timesteps):
43:     stokes_solver.solve()  # solving Stokes
44:     energy_solver.solve()  # solving energy
45:     checkpoint_file.save_function(u, name="Velocity", idx=timestep)  # Store velocity (uobs)
46:
47: checkpoint_file.save_function(T, name="Temperature", idx=max_timesteps - 1)  # Store final temperature (Tobs)
48: checkpoint_file.close()
```

**Listing 1.** Selected lines from G-ADOPT code, demonstrating generation of our reference isoviscous simulation in an enclosed square domain.

cooled from above ($T_S = 0$). The initial condition is generated by a Gaussian anomaly of amplitude $0.1$, centred at $x_0 = (0.5, 0.2)$, superimposed on top of an average temperature profile generated by two error functions, representing top and bottom thermal boundary layers. Starting from this initial condition, the model is run for 80 time steps of $\delta t = 4 \times 10^{-6}$: the

time required for the temperature anomaly to form a plume that reaches the domain's top boundary.

Listing 1 shows selected lines of a G-ADOPT script used to generate this reference synthetic experiment. The first step, illustrated on line 1, is to import the G-ADOPT module (which provides access to Firedrake and associated functionality). We next need a mesh, for which we use a built-in Firedrake meshing function. The computational domain is a unit square with $150 \times 150$ elements, loaded on lines 4-6. The function spaces, within which our solutions are defined, are specified as follows:

1. A vector function space, v, is specified for the velocity field (line 9), employing a $Q2$ discretisation.

   2. A scalar function space, w, is specified for pressure (line 10), utilising a $Q1$ discretisation.

   3. These are combined on line 11 to create the mixed function space, z, for the Stokes (velocity and pressure) system.

   4. A function space Q, is specified for the temperature field (line 12), using a $Q2$ discretisation.





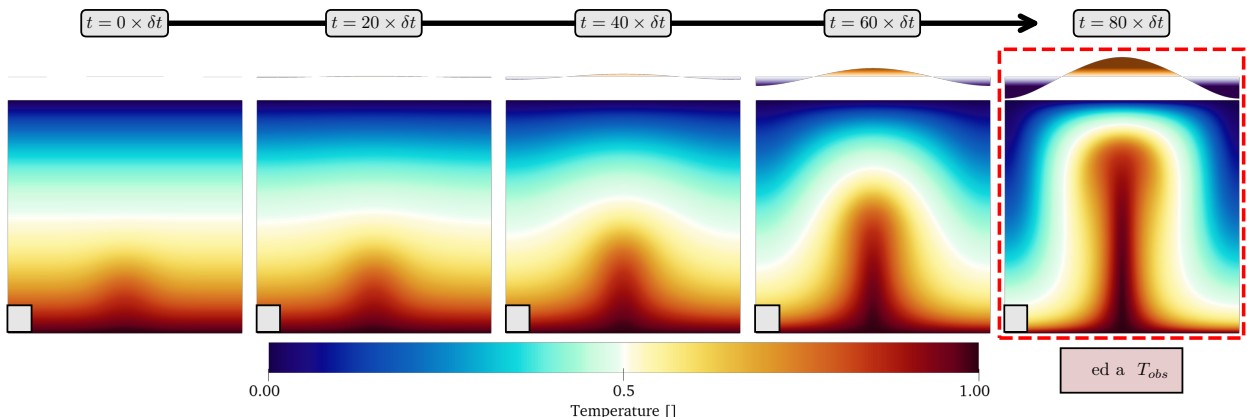

**Figure 2.** Reference forward simulation: The initial condition is generated by superimposing a gaussian anomaly of maximum amplitude 0.1 centred at $x_0 = (0.5, 0.2)$ on top of an adiabatic profile. The simulation runs forward in time from this initial condition, until the plume-like feature approaches the top boundary after 80 times-steps. Also visualised at top of each snapshot are the normal stresses acting on the top boundary, which are proportional to dynamic topography. Boxed in dashed red is the final state of the simulation that is subsequently used as $T_{obs}$ in our synthetic adjoint simulation.

We specify functions to hold our solutions on lines 14-18. The temperature field, `T`, is initialised on line 20 using a symbolic

expression for the coordinates from line 18. The initialisation includes a 1-D profile along the y-axis (line 21) and a Gaussian anomaly, as specified above (line 22). This problem has a constant pressure nullspace, defined as `Z_nullspace` on line 24, which will subsequently be passed to the solver, and PETSc will seek a solution in the space orthogonal to the provided nullspace. A checkpoint file is initiated on lines 26-27 to retain the necessary fields for the subsequent adjoint inversion. Important constants in this problem (Rayleigh Number, $Ra$, time-stepping parameters, `delta_t` and `max_timesteps`) are defined on lines 28-31, with

the Boussinesq approximation specified on line 29 (later passed on to the Stokes and energy systems to determine which terms are to be assembled). Boundary conditions for velocity and temperature are specified on lines 34 and 35, respectively. The latter uses integer mesh markers to tag entities of meshes, with boundaries tagged as follows: tag 1 corresponds to the plane $x = 0$ (left); 2 to $x = 1$ (right); `"bottom"` to $y = 0$; and `"top"` to $y = 1$.

We now solve the variational problem, with solver objects for the energy, `energy_solver`, and Stokes, `stokes_solver`, systems

created on lines 37 and 38. For the energy system we pass in the solution field T, velocity $u$, the physical approximation, time step, temporal discretisation approach (i.e. implicit middle point) and boundary conditions. For the Stokes system, we pass in the solution fields `z`, Temperature, the physical approximation, boundary conditions and the nullspace object. Solution of the two variational problems is undertaken by PETSc. The time-loop is defined on lines 42-45, with the Stokes system solved on line 43, the energy equation on line 44, and velocities (for later use as time-dependent surface constraints in our adjoint

inversions) checkpointed on line 45. Figs. 2-A to E show temporal snapshots of the reference forward simulation. We note that the final temperature field (Fig. 2-E), subsequently utilised in the adjoint inversion, is also checkpointed on line 47.



### 3.1.2 Inverse Problem

Only trivial changes are required to convert the forward problem outlined in Listing 1 into its corresponding adjoint, which are outlined in Listing 2. We first augment our imports with `gadopt.inverse` (line 2): this provides access to crucial Dolfin-Adjoint functionalities harnessed by Firedrake, enabling overloading differentiation and taping of finite element operations. On line 4, we activate disk checkpointing of intermediate forward solutions, ensuring that these fields — otherwise retained in memory — are available as inputs for solving the adjoint equations. For this problem, we select the initial temperature as the control, symbolised as `Tic` on line 7, using a $Q1$ function space defined on line 6. On line 8, we define the average temperature field for regularisation terms using the same $Q1$ function space. Despite utilising $Q2$ elements for the forward temperature computation, opting for a basis function with a lower polynomial degree for the control is advantageous, as it curtails computational expense during the internal optimisation algorithm's operations by reducing the number of degrees of freedom in the solution. Furthermore, using a lower polynomial degree reduces the complexity and regularisation requirements of the optimisation problem, helping to avoid over-fitting of the solution (e.g., Hastie et al., 2009).

On line 10, a checkpoint file is opened. The reference final temperature field is subsequently loaded on line 11, with our guess at the initial temperature condition specified on line 12 (noting that it corresponds to the terminal temperature field of the reference simulation). We specify the control for Dolfin-Adjoint on line 15, followed by projection of the initial temperature condition onto `T` on line 18. During execution of the time loop (lines 20-24), we solve Stokes and energy systems, after which we load velocities from the reference simulation on line 24, used when accumulating contributions to the surface velocity misfit, `u_misfit`, on line 24. After the time loop, we define several components of the objective functional. Specifically, we establish the damping term and its associated normalisation factor (lines 29-31), the smoothing term and associated normalisation (lines 32-33), two normalisation terms associated with final state temperature and surface velocity misfits (lines 34-35), and the misfit associated with the final temperature field (line 36). These terms are combined on lines 38-42 as our `objective`, using weights ($\alpha_u$, $\alpha_d$, and $\alpha_s$) specified on line 37, and is later utilised on line 44 to define the reduced functional. We note that values for $\alpha_u$, $\alpha_d$, and $\alpha_s$ are systematically tested herein.

### 3.1.3 Investigating the Derivative

Our *initial guess* for $T_{IC}$ is set to the final *'observed'* temperature field $T_{obs}$ (i.e., the terminal temperature field of the forward model). This choice is grounded in the findings of Horbach et al. (2014), and our own tests, which demonstrate that the minimisation problem possesses a strong minimum, rendering it insensitive to the initial guess. The first optimisation iteration starts from this initial guess (Fig. 3-A) and runs forward in time to arrive at the first modelled terminal temperature field $T_{t=t_F}$ (Fig. 3-B). Compared to the reference $T_{obs}$ (Fig. 3-C), the model is further advanced in time, with the plume tail in the lower mantle narrower, hot buoyant anomalies spread throughout the upper part of the domain, generating a cold return flow and resulting in thinning of the thermal boundary layers. Previously, we noted that for our reconstruction simulations, the objective functional, denoted as $J$, encompasses four distinct terms: (i) final temperature field; (ii) surface velocity misfit; (iii) smoothing terms; and (iv) damping terms. The relevance of the three latter terms is gauged by their respective weighting





```
 1: ...
 2: from gadopt.inverse import *
 3: ...
 4: enable_disk_checkpointing()   # Enable checkpointing to disk for adjoint
 5: ...
 6: Q1 = FunctionSpace(mesh, "CG", 1)  # Control (Tic) function space
 7: Tic = Function(Q1, name="Initial Temperature")    # Control: Initial temperature
 8: Taverage = Function(Q1, name="Average Temperature")  # Average temperature
 9:
10: checkpoint_file = CheckpointFile("Checkpoint_State.h5", "r")   # File for loading reference constraints
11: Tobs = checkpoint_file.load_function(mesh, "Temperature", idx=max_timesteps - 1)  # Load reference final state
12: # Initialise control to reference final T
13: Tic.project(checkpoint_file.load_function(mesh, "Temperature", idx=max_timesteps - 1))
14: ...
15: control = Control(Tic)   # definition of control
16: T.project(Tic, bcs=energy_solver.strong_bcs)   # projection from Q1 to Q2,
17:                                                 # and imposing boundary conds
18: # Populate the tape by running the forward simulation
19: # and calculate surface velocity misfit
20: for timestep in range(0, max_timesteps):
21:     stokes_solver.solve()
22:     energy_solver.solve()
23:     uobs = checkpoint_file.load_function(mesh, name="Velocity", idx=timestep)  # load uobs
24:     u_misfit += assemble(dot(u - uobs, u - uobs) * ds_t)   # accumulate misfit
25:
26: checkpoint_file.close()
27:
28: # Define the objective functional
29: damping_mask = gaussian(X[1], 1.0, 0.1) + gaussian(X[1], 0.0, 0.1)
30: damping = assemble(damping_mask * (Tic - Taverage) ** 2 * dx)
31: norm_damping = assemble(damping_mask * Taverage**2 * dx)
32: smoothing = assemble(dot(grad(Tic - Taverage), grad(Tic - Taverage)) * dx)   # smoothing term
33: norm_smoothing = assemble(dot(grad(Tobs), grad(Tobs)) * dx)   # normalisation for smoothing
34: norm_obs = assemble(Tobs**2 * dx)   # normalisation for Tobs
35: norm_u_surface = assemble(dot(uobs, uobs) * ds_t)   # normalisation for u surface
36: t_misfit = assemble((T - Tobs) ** 2 * dx)   # Tobs misfit
37: alpha_u, alpha_d, alpha_s = 0.1, 0.1, 0.1
38: objective = (
39:     t_misfit
40:     + alpha_u * (norm_obs * u_misfit / max_timesteps / norm_u_surface)
41:     + alpha_d * (norm_obs * damping / norm_damping)
42:     + alpha_s * (norm_obs * smoothing / norm_smoothing))
43: # Define the reduced functional
44: reduced_functional = ReducedFunctional(objective, control)
```

**Listing 2.** Changes compared to the forward script in Listing 1 required to define the reduced functional used in the adjoint problem. The three `alpha` parameters on line 36 denote the three weighting parameters for surface, damping and smoothing terms, respectively.

factors, namely, $\alpha_u$, $\alpha_s$, and $\alpha_d$. When combining terms linearly in the objective functional, the superposition principle dictates that the gradient of the entire objective functional is essentially the cumulative sum of the gradients of its constituent terms in $J$. Consequently, it becomes instructive to visualise and validate the gradient of each individual term within the misfit functional.

Fig. 3-D to G display the gradient fields corresponding to the four terms in the objective functional. The gradient for the final temperature misfit is presented in Fig. 3-D: it implies that a better match to the final state temperature field is achievable

through changes to the initial condition that include a major reduction in temperature in the domain's centre, complemented by an increase in temperature moving towards the domain's edges, particularly towards the domain's upper regions. In Fig. 3-E, we illustrate the gradient of the cumulative surface velocity misfit with respect to $T_{IC}$: it reveals sensitivities extending to the domain's base, indicating that to better align with the 'observed' surface velocities, an optimal $T_{IC}$ should be decreased in the middle, but increased towards the domain's boundaries. For the smoothing term (Fig. 3-F), the highest values emerge in

areas with the most abrupt changes in $T_{IC}$. Given our optimisation strategy works by moving towards corrections opposite to the gradient's direction, this subsequently results in iterative refinement and smoothing of the solution. For the damping term (Fig. 3-G), the gradient aims to minimise fluctuations of the field in relation to the ambient temperature profile ($\bar{T}$), adjacent to the top and bottom boundaries, implying that this can be achieved by reducing boundary layer temperatures in the centre but increasing boundary layer temperatures towards the domain's sides.





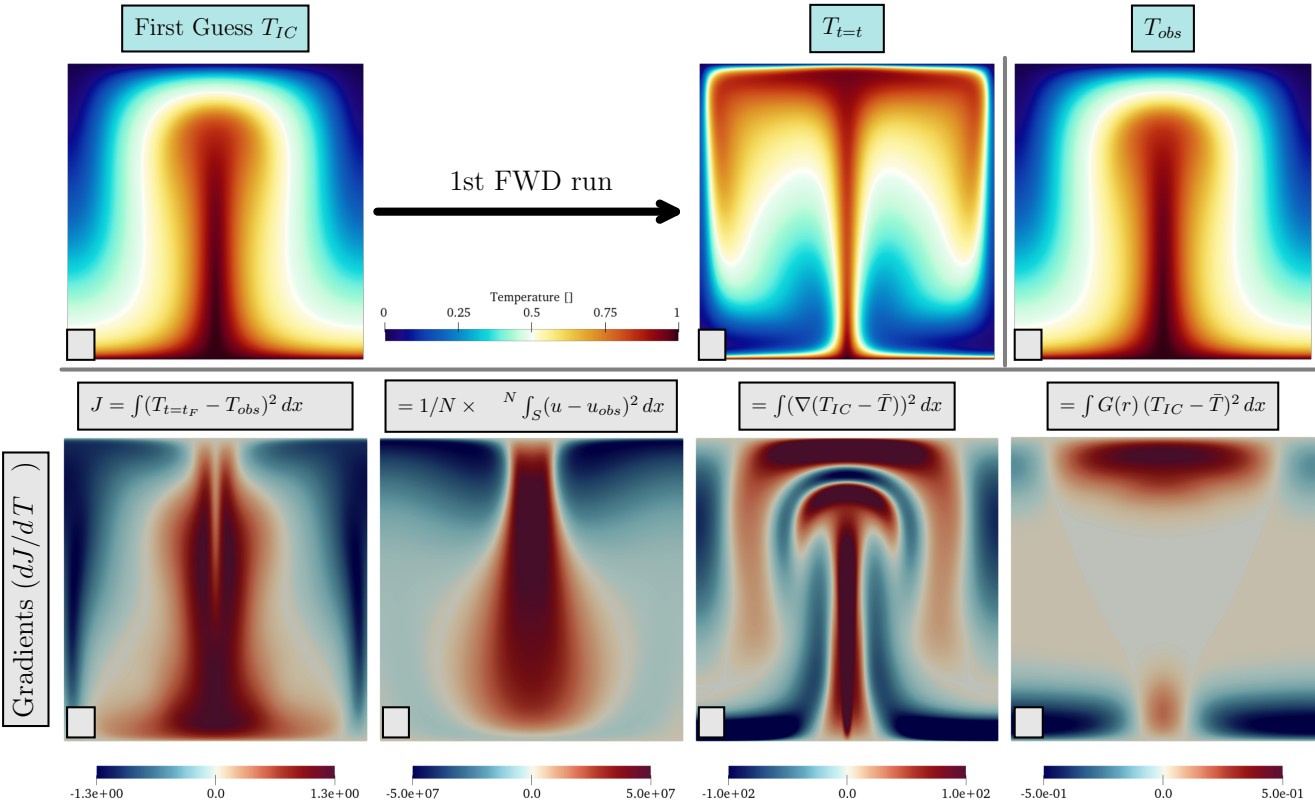

**Figure 3.** First forward run and associated gradients: **(A)** is the first guess for $T_{IC}$. For all experiments in this study we choose the *'observed'* temperature field (i.e., the final state) as the initial guess for our optimisations. **(B)** is the final temperature field after integrating forward in time from (A) for 80 time steps. **(C)** is the reference final temperature field, $T_{obs}$, which is used in the definition of the misfit functional. **(D)**-**(G)** illustrate the gradient fields. **(D)** is gradient of the final temperature misfit. **(E)** is the gradient of the total surface velocity misfit. **(F)** and **(G)** are for the regularising smoothing and damping terms, respectively.

### 3.1.4 Verification of Gradients: Taylor Remainder Convergence Test

A fundamental tool used in verification of gradients is the Taylor remainder convergence test (Farrell et al., 2013b). For the reduced functional, $J(T_{IC})$ defined in Eq. 6 and its derivative $\frac{\mathrm{d}J}{\mathrm{d}T_{IC}}$, it can be proven that,

$$|J(T_{IC} + h\,\delta T_{IC}) - J(T_{IC}) - h\,\frac{\mathrm{d}J}{\mathrm{d}T_{IC}} \cdot \delta T_{IC}| \longrightarrow 0 \text{ at } O(h^2). \tag{33}$$

The expression on the left hand side of Eq. 33 is termed the second-order Taylor remainder. This term's convergence rate of $O(h^2)$ serves as a strong foundation for verifying any computational implementation meant for determining $\frac{\mathrm{d}J}{\mathrm{d}T_{IC}}$ (the adjoint code) with respect to a specific functional that computes $J(T_{IC})$ (the forward code). Given any arbitrary selection of $h$ and $\delta T_{IC}$, halving the value of $h$ should decrease the magnitude of the second-order Taylor remainder by a factor of 4. Grounded in this theoretical prediction, we employ these so-called *Taylor tests* to confirm the accuracy of the determined gradients.





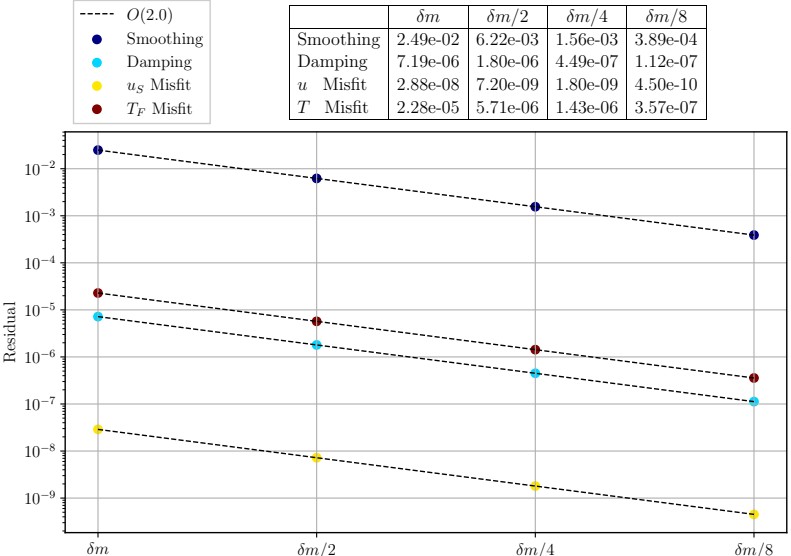

**Figure 4.** Second-order Taylor Remainder Test: For each gradient field, the test is performed by computing the functional and the associated gradient when randomly perturbing the initial temperature field $T_{IC}$ and subsequently dividing the perturbations by a factor of two at each level. The dashed line is the theoretical convergence rate of $O(2.0)$.

We perform a second-order Taylor remainder test for each term of the objective functional. Fig. 4 shows the results: in each case, the gradient fields is calculated for random perturbations of the initial temperature field $T_{IC}$ and subsequent halving the amplitude of the perturbations (1/2, 1/4, 1/8). All four Taylor remainder tests show an $O(2.0)$ convergence rate, consistent with theoretical expectations.

### 3.1.5 Efficiency

Another metric that can be used to assess the suitability of our framework for large-scale mantle convection optimisation problems is the efficiency of derivative calculations. Given the iterative nature of our inverse problem, where derivatives are computed frequently, any efficiency gain, or the lack thereof, can have profound implications for the overall computational cost and feasibility of our automated approach. The computational efficiency can be measured by comparing the computational time of a derivative calculation to that of a forward calculation. Using this reference, a theoretical optimum is defined which measures the ratio of the time that is required to calculate one set of forward and adjoint calculations to one forward calculation. For the Stokes problem we have detailed, in which a linear rheology has been employed, this ratio is considered to be 2.0 (Naumann, 2011; Funke and Farrell, 2013). This is primarily due to the similarity of the forward and adjoint systems. For the simulations presented in this section, we achieve a ratio of 2.01, consistent with theoretical expectations, thus demonstrating the efficiency of our approach.





```
1: ...
2: # Perform a bounded nonlinear optimisation
3: T_lb = Function(Tic.function_space(), name="Lower bound temperature").assign(0.0)
4: T_ub = Function(Tic.function_space(), name="Upper bound temperature").assign(1.0)
5:
6: minimisation_problem = MinimizationProblem(reduced_functional, bounds=(T_lb, T_ub))
7:
8: optimiser = LinMoreOptimiser(minimisation_problem, minimisation_parameters, checkpoint_dir="
       optimisation_checkpoint")
9: optimiser.run()
```

**Listing 3.** Necessary changes to solve the minimisation problem. `T_lb` and `T_ub` are the lower and upper bounds for the minimisation problem, respectively.

### 3.1.6 Optimisation

Executing an optimisation task with G-ADOPT is straightforward. Once the reduced functional is set up (see Listing 2), only a few additional lines of Python are required (see Listing 3). As we use a bounded method for our optimisation problem, we specify a set of upper and lower bounds for the algorithm, on lines 3-4. Subsequently, a minimisation problem is outlined (line 6), using both the reduced functional and the designated bounds. This minimisation problem, together with the associated parameters for optimisation, are passed to the Lin-More optimisation algorithm in ROL (line 8), which is executed on line 9.

Using this framework, we perform a suite of 81 different inverse simulations that aim to find the most optimal combination of the three weightings ($\alpha_u$, $\alpha_d$, and $\alpha_s$) that results in the best solution for $T_{IC}$ when compared to the reference initial temperature field. The simulations are obtained by sweeping through values in ranges of $[10^{-1}, 10^{-3}]$, $[10^{-2}, 10^{-4}]$ and $[10^1, 10^{-7}]$ for $\alpha_d$, $\alpha_s$ and $\alpha_u$, respectively.

Fig. 5 provides an overview of the outcomes from a subset (16 out of 81) of these optimisation exercises. Minimisation of 520 the objective functional is shown in Fig. 5-A, alongside two additional metrics: (i) the misfit between the reconstructed final temperature field and $T_{obs}$, termed the *final misfit* (Fig. 5-B); and (ii) the misfit between the reconstructed initial condition, $T_{IC}$, and the reference initial condition, highlighting the quality of the reconstructed initial condition (Fig. 5-C). The reduction in the metrics in all cases is reported versus the cost, which is the sum of the number of forward and adjoint calculations. A consistent pattern is observed across all reconstruction simulations for these three metrics. Firstly, the solutions are unique, as 525 all converge to a consistent initial condition following roughly 100 iterations (a cost of 200 forward and adjoint calculations). However, the trajectory to this solution varies based on the smoothing weight, denoted by $\alpha_s$. A significant portion of the simulations with $\alpha_s = 10^{-1}$ exhibit subpar performance in the initial stages, due to over-smoothing, with most of the best performing simulations utilising $\alpha_s = 10^{-2}$. Nevertheless, despite differences in convergence rates, all simulations eventually converge to similar misfits in all three metrics.

Fig. 6 showcases multiple iterations from the best-performing simulation using $\alpha_u = 10^{-2}$, $\alpha_s = 10^{-3}$, and $\alpha_d = 10^{-2}$. As mentioned previously, the reference final condition ( Fig. 6-B) is used as our starting guess for the inverse simulation (Fig. 6-C), which yields substantial differences in the modelled final state (Fig. 6-K), as illustrated through the squared difference between reconstructed and reference temperature (Fig. 6-O). Leveraging derivative information, the inverse simulation corrects the initial condition through an iterative approach. The initial conditions obtained after 20, 50, and 100 iterations are shown in 535 Figs. 6-D, E, and F, respectively. These solutions reveals that most corrections occur during the initial iterations, with significant



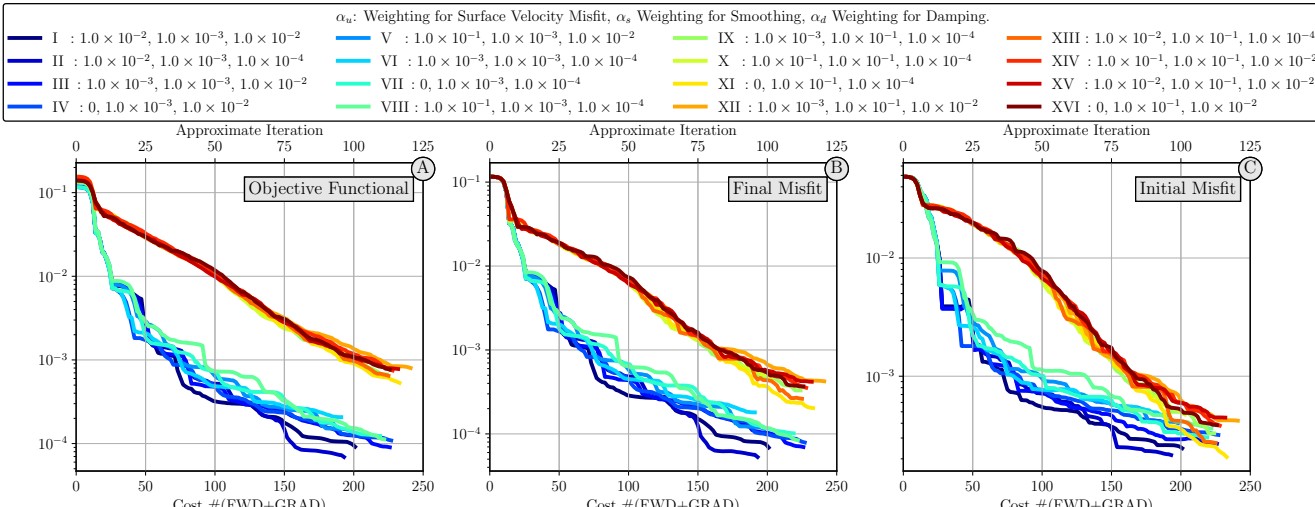

**Figure 5.** An overview of optimisation results across all experiments: (A) Illustrates the minimisation of the objective functional. (B) Depicts the final misfit, representing the difference between the reconstructed final temperature field and $T_{obs}$. (C) Highlights the initial misfit, characterising the discrepancy between the reconstructed initial condition $T_{IC}$ and the reference initial condition. The bottom horizontal axis in all figures reports the cost, here defined as the sum of forward and derivative calculations. The top axis shows the approximate equivalent number of iterations, which simply is the cost divided by two. Across all metrics, simulations exhibit a consistent pattern: they converge to the same solution approximately after 100 iterations, though the path to this solution diverges depending on the smoothing weight, $\alpha_s$.

improvement in the domain's upper part by iteration 20, albeit with some remaining noise in the solution (see misfit in Fig. 6-H). By iteration 100, the solution (visible in Fig. 6-F and N) closely mirrors the reference initial condition, represented by misfit values that have diminished by three orders of magnitude (Fig. 5-B and C).

Fig. 7 compares the evolution of reference forward (A-E) and reconstructed simulations (F-J), with differences highlighted
(K-O). Furthermore, surface normal stresses are displayed as indicators of surface dynamic topography. For the reconstructed simulation (Fig. 7-F), the initial condition is well captured as a Gaussian anomaly in the domain's lower section, superimposed on a depth-dependent field, similar to the reference initial condition. This initial temperature anomaly ascends, culminating at the surface after 80 time steps. The precision of the reconstruction is evident from the misfit panels, with negligible differences between reference and reconstructed simulations throughout the model's evolution (less than 0.01). This is further substantiated
by a reduction, by over four orders of magnitude, in the objective functional for both initial and end states (Fig. 5-B and C). The accuracy of reconstructed surface normal stresses are evidenced by negligible discrepancies in the associated misfit visualisation.





| Parameter | Symbol | Value |
|---|---|---|
| Viscosity [] | $\mu$ | — |
| Plastic viscosity [] | $\mu_p$ | — |
| Linear (temperature-dependent) viscosity [] | $\mu_{lin}$ | — |
| Principal strain rate tensor | $\epsilon_i i$ | — |
| Ambient viscosity | $\mu(r)$ | — |
| Yield stress (at Earth's surface) | $\sigma_{yield}$ | $2 \times 10^4$ |
| Minimum plastic viscosity | $\mu^*$ | 0.1 |
| Temperature dependence of viscosity | $\Delta_T$ | $ln(80)$ |

**Table 2.** Parameters employed for Earth-like twin experiment scenarios.

## 3.2 An Earth-like Problem in an Annulus Using a Visco-plastic Rheology

We next analyse a set of reconstructions which utilise a reference twin that more closely mimics Earth's geometrical and
rheological characteristics. We use a 2-D annular domain, generated by extruding 128 times radially from a circular manifold
consisting of 512 cells. Inner and outer radii are set to $1.22$ and $2.22$ respectively, ensuring unit depth and maintaining a
comparable ratio between surface and CMB radii as Earth's mantle. We set $Ra = 10^7$ and adopt a composite visco-plastic
rheology, with effective viscosity determined via a harmonic mean, represented as:

$$\frac{1}{\mu} = \frac{1}{\mu_p} + \frac{1}{\mu_{lin}}, \qquad\qquad \mu_p = \mu^* + \frac{\sigma_{yield}}{\epsilon_{ii}}, \qquad\qquad \mu_{lin} = \mu_0(r)\exp\left(\frac{\Delta_{\mu,T}}{T}\right). \qquad (34)$$

Parameters specified in Eq. 34 are listed in Table 2. As demonstrated by Davies et al. (2022), the changes necessary to trans-
form our forward model from a square to an annular domain, and from an isoviscous to a visco-plastic rheology, are only
minor, noting the Firedrake has already been validated for simulations of this nature (see Davies et al. (2022) and repository
accompanying this paper for a comprehensive script).

To determine the reference simulation's initial condition, we commence from a starting state with small-scale spherical
harmonic perturbations and advance forward in time until a quasi-steady state is achieved (i.e., when basal and surface heat-
fluxes are roughly in balance). The resulting state (Figs.8-1.A and 2.A), which contains Earth-like subduction zones that
descend from the upper thermal boundary layer and thermal plumes that rise from the lower thermal boundary layer, forms our
reference initial condition. We subsequently advance forward for a duration of $t = 200 \times (5 \times 10^{-6})$ to reach the reference final
state shown in Figs. 8-1.G. The sequence presented in Figs. 8-1.A-G and 2.A-G trace the temporal development of reference
temperature and viscosity fields, respectively. The viscosity field spans approximately three orders of magnitude, extending
from low asthenospheric values of $\approx 1.0$, rising to $140$ within the cooler segments of the lower mantle, and decreasing to $0.4$
in locations of high strain-rates at surface convergent boundaries.





### 3.2.1 Verification of Gradients

As with the previous case examined, reconstruction simulations are conducted with an objective functional encompassing the
misfit component related to the terminal temperature field, accumulative surface velocity misfits, and regularisation terms. For
consistency with the previous case, we first confirm the accuracy of the calculated gradient fields corresponding to each com-
ponent in the objective functional by performing second-order Taylor remainder convergence tests. The convergence outcomes
for these gradient distributions are illustrated in Fig. 9, demonstrating convergence of $O(2.0)$, as expected.

### 3.2.2 Efficiency

In the experiments detailed in this section, an efficiency of $1.45$ is achieved. This exceeds the previously outlined theoretical
efficiency of 2.0 for the isoviscous Stokes model in Sec. 3.1.5, due to a major difference in the forward and adjoint momentum
equations: while the forward momentum equation employs a non-linear visco-plastic rheology and requires multiple linear
Newton solves per time-step, the adjoint momentum remains linear. The forward model is therefore more computationally
demanding, explaining the improved efficiency ratio.

### 3.2.3 Optimisation

As highlighted with our previous example, absolute and relative variations in weighting of different objective functional com-
ponents can generate solutions with distinct properties, some of which provide an improved match to the reference simulation.
Accordingly, it is vital to assess the consequences of these distinct weight combinations for the case considered here. To address
this, we have undertaken 21 simulations, adjusting the parameters $\alpha_u$, $\alpha_s$, and $\alpha_d$ within the intervals $[0.05, 0.1]$, $[0.01, 0.1]$,
and $[0.01, 0.1]$ respectively, with values motivated by the results of our previous set of simulations. The collective convergence
of all 21 simulations is illustrated in Fig. 10.

Our objective functional (Fig. 10-A), has initial values that range between $\sim 1 \times 10^{-1}$ and $2 \times 10^{-1}$. We consistently observe
a reduction of an order of magnitude or more in this measure. Notably, the steepest decline is seen within the initial 50 iterations
(cost $\sim 100$). The simulations denoted as X, XI exhibit the largest reduction, with a consistent reduction trajectory even when
approaching iteration 200 (cost $\sim 400$).

The final temperature misfit (Fig. 10-B) exhibits different trends to the objective functional. In the initial iterations, the
largest reductions in final misfit are observed for simulations I and II, with simulation XII trailing behind. Despite displaying
a lower misfit reduction up until iterations 140 and 180 (cost $\sim 280 - 360$) simulations X and XI eventually display a similar
misfit by iteration 200 (cost $\sim 400$). When we turn to reduction in the initial misfit (Fig. 10-C), an entirely different trend
comes to light: simulations I and II sustain their reduction until around iteration 150 (cost $\sim 300$), after which they plateau.
Conversely, many other simulations plateau at misfit values that are, on average, twice as high.

In our analysis, the reconstruction quality is predominantly governed by three key weighting parameters: surface velocity
misfit ($\alpha_u$), smoothing ($\alpha_s$), and damping ($\alpha_d$). These parameters calibrate the significance of different objective terms in
relation to the final temperature misfit term. Thus, the primary metric to assess a reconstruction is the simultaneous reduction





of misfits for both initial and final states. Incorporating the misfit associated with surface velocities enhances the quality of the reconstructions, noting that higher weightings of the surface velocities require higher values of smoothing. Among the weighting parameters, $\alpha_s$ shows to have the highest effect in convergence outcomes: higher values for $\alpha_s$ lead to over-regularisation, thereby limiting the role of sensitivity information tied to misfit terms in the solution. $\alpha_d$ offers a more varied range of effective values, which aligns with its role confined to thermal boundary layers and, consequently, its lesser impact on

the overall numerical domain. Therefore, Case I, characterised by $(\alpha_u, \alpha_s, \alpha_d) = (1 \times 10^{-1}, 1 \times 10^{-1}, 1 \times 10^{-2})$, emerges as the optimal set of weighting parameters for our reconstructions.

In contrast to the three orders of magnitude reduction in the misfit functions from the isoviscous experiment, our non-linear experiment exhibits a modest reduction of $O(1)$. The key factors contributing to this are the one-order-of-magnitude higher Rayleigh number and the extended total simulation time for the non-linear experiment. Representing a far longer period of

reconstruction, these factors imply more loss of information during the inversion process. Additionally, while the isoviscous experiment focused on a single temperature anomaly, this simulation tackles whole-mantle convection, with numerous, and occasionally complex and highly time-dependent, anomalies, reflecting the more intricate visco-plastic rheology. Nonetheless, this order of magnitude reduction in misfit translates into a satisfactory reconstruction of the initial condition, demonstrating the efficacy and robustness of the numerical approaches employed.

This is confirmed by visual inspection of the best reconstruction model, with temperature, viscosity and surface normal stresses presented and compared to the reference case in Fig. 11 (marked case I in Fig. 10, using values of $\alpha_u = 10^{-1}$, $\alpha_s = 10^{-1}$ and $\alpha_d = 10^{-2}$). At $t = 0$, the reconstructed temperature field exhibits upwelling and downwelling features that are reconstructed in the correct locations, although temperature anomalies are generally smoother than those of the reference case (Fig. 11-1.A). The corresponding viscosity field mirrors this smoothness, despite capturing weaker convergence zones at the surface. Despite these variations, the spatial misfit is generally below $10^{-2}$, (Fig. 11-5.A) with errors over $0.05$ restricted

to sharper features that are inevitably smoothed in the reconstruction process. This smoothness is also reflected in recovered surface normal stresses, with highs and lows correctly positioned, albeit at longer wavelengths than the reference case.

Given the application of a free-slip boundary condition at the surface of this simulation, a prominent outcome of this set of experiments is the emergence of sharp subducting slabs and weak zones at the top boundary as the simulation evolves

(Figs. 11-3.B and 4.B). The marked decrease in misfit over time (Fig. 11-5.B) confirms the development of more detailed convective patterns. As the simulation evolves, reconstructed plume features become more precise and reconstructed surface normal stresses more closely resembles the reference case. This enhancement progresses up to the final time-step, where the reconstructed thermal field and surface normal stresses are indistinguishable from those in the reference simulation, reflected via an order of magnitude reduction in the spatial misfit field.

**4 Discussion**

Robust reconstructions of the spatial and temporal evolution of Earth's mantle and its diverse surface expressions is critical to scientific progress across the geosciences. It requires the construction of a digital twin: a vital instrument for analysing and



revealing the complex interplay between the mantle and Earth's other systems. To this end, the adjoint method provides the necessary means for obtaining and analysing model sensitivities with respect to earlier mantle states. A burgeoning number of studies exploiting this methodology for reconstructions of mantle convection have emerged in recent years (e.g., Bunge et al., 2003; Ismail-Zadeh et al., 2004; Liu et al., 2010; Spasojevic et al., 2009; Li et al., 2017; Price and Davies, 2018; Ghelichkhan et al., 2021). Nevertheless, the derivation, implementation, and validation of adjoint systems for coupled, non-linear, time-dependent systems remains notoriously difficult. It is due to these difficulties that existing applications of the geodynamic adjoint method often include major simplifications, either incorporating an oversimplified treatment of mantle rheology (e.g., Colli et al., 2018; Ghelichkhan et al., 2021), neglecting certain (coupling) terms in the adjoint equations (e.g., Ismail-Zadeh et al., 2004), or both (e.g., Liu and Gurnis, 2010): they are therefore likely limited in their applicability to realistic Earth scenarios. In this study, we leverage the latest advances in scientific computing to overcome these limitations and develop G-ADOPT, an open-source numerical framework for geoscientific adjoint reconstructions, developed in full compliance with FAIR (Findable, Accessible, Interoperable, Reusable) principles (Wilkinson et al., 2016).

G-ADOPT is underpinned by three primary software elements. The first is Firedrake (Ham et al., 2023), an automated system for solving partial differential equations using the finite-element method. In our previous work (Davies et al., 2022) we examined the applicability of Firedrake for geodynamical simulations, confirming its accuracy, efficiency, extensibility and parallel scalability through comprehensive benchmarks and state-of-the-art mantle convection simulations. The second element is Dolfin-Adjoint (Farrell et al., 2013a; Mitusch et al., 2019), a system that automatically generates the discrete adjoint from forward models designed in Firedrake. Dolfin-Adjoint elevates the conventional abstraction of automatic differentiation from individual floating point operations to complete systems of differential equations, leveraging the high-level mathematical abstraction of finite element problems and their symbolic representation in UFL (Alnæs et al., 2014). The adjoint systems derived by Dolfin-Adjoint are UFL expressions and valid Firedrake inputs. Therefore, they inherit the parallel support native to the forward model, which results in optimal computational efficiency. The third element is the Rapid Optimisation Library, ROL, a Trilinos package for large-scale optimisation problems (The ROL Project Team, 2022), enhanced herein with intra-optimisation checkpointing functionality.

We have demonstrated the applicability of G-ADOPT for time-dependent geodynamic reconstructions herein. The objective functional utilised in our reconstructions is composed of two distinct misfit components. The first is a term that quantifies the misfit corresponding to the observed final state temperature field, analogous to the present-day temperature field within Earth's mantle as obtained through a combination of mantle mineralogical models (e.g. Chust et al., 2017; Stixrude and Lithgow-Bertelloni, 2011) and seismic imaging (e.g., Rawlinson et al., 2010; French and Romanowicz, 2014; Simmons et al., 2015; Bozdağ et al., 2016; Koelemeijer et al., 2016; Fichtner et al., 2018). The second term corresponds to observed surface velocities, accessible through plate tectonic reconstruction models (e.g., Müller et al., 2019). Additionally, smoothing and damping terms have been incorporated to enforce regularity in our solutions.

Our study analysed two sets of reconstructions of systematically increasing complexity. We first examined the evolution of a single ascending hot anomaly in an enclosed isoviscous square domain. By taking advantage of the simplicity of the geometry and rheological properties, we were able to deliver an in-depth examination of the gradients for each term, including





a parameter-space search to ascertain optimal weighting parameters. Our results reveal a general convergence of the solutions, notwithstanding substantial variations in convergence rates subject to the weightings. Additionally, although not detailed in this

paper, we have explored a number of different optimisation methods and parameters. Through this comprehensive analysis, we are confident that the problem possesses a stable solution that can be found through an appropriate combination of weighting parameters. The second set of reconstruction experiments explored convection with a stress-, depth- and temperature-dependent rheology at the convective vigour of Earth's mantle, demonstrating the feasibility of reconstruction studies for Earth's mantle with a non-linear rheology. The weightings selected for this series of experiments were broadly consistent with the first set.

Given the success at reproducing surface velocities and normal stresses, our findings suggest that reconstruction models of Earth's mantle can serve as a powerful means for probing changes in the landscape at Earth's surface induced by mantle dynamics (e.g., Friedrich et al., 2018; Hoggard et al., 2021; Davies et al., 2023).

In both experimental sets, we assessed the numerical efficiency of our framework by evaluating the cost ratio between forward and adjoint calculations. In the first set, our results produced a ratio of 2.01, aligning with the theoretical efficiency of

2.0 (e.g. Naumann, 2011). In the second set, where we solved the nonlinear forward equations, we observed a ratio of 1.45. This efficiency is attributed to the linearised nature of the adjoint method: even when applying nonlinear rheologies in the forward equations, the adjoint equations remain linear. We also conducted second-order Taylor remainder convergence tests for each of the objective functional terms to validate the adjoint calculations. Our assessments demonstrate the accuracy of the derivative calculations (Figs.4 and 9). These Taylor remainder convergence tests provide a robust basis for future validations

of geodynamic adjoint frameworks.

Our experiments incorporate two significant simplifications relative to realistic-Earth scenarios, which were necessary to facilitate the number of reconstruction simulations analysed: (i) the use of a 2-D computational domain; and (ii) application of the Boussinesq approximation instead of more pertinent approximations such as anelastic-liquid approximations (e.g., Jarvis and McKenzie, 1980). Nevertheless, the composable nature of G-ADOPT should alleviate any concerns regarding the extensi-

bility of our framework to these more realistic problem sets. Our prior work in Davies et al. (2022) demonstrates the flexible nature of our approach: for example, transitioning our 2-D annulus simulations to a 3-D spherical shell domain can be achieved via changes to only a few lines of Python. The application of G-ADOPT for reconstructing Earth's mantle evolution using non-linear rheologies and compressiblity will be the subject of future investigations, although we note that the forward modelling approach has already been developed (Davies et al., 2022). Moreover, our framework is extensible to various other problems

in geodynamics. These include utilising principal stress directions (e.g., Reuber et al., 2020), surface plate velocities (e.g., Ratnaswamy et al., 2015; Bocher et al., 2018), and/or residual depth measurements (e.g., Panasyuk and Hager, 2000; Spasojevic et al., 2009) to explore the mantle's rheological properties, and to study the visco-elastic adjustment of Earth's surface in response to the melting of Earth's polar ice-sheets (Al-Attar and Tromp, 2014; Martinec et al., 2015, e.g.,), and post-seismic deformation following significant subduction earthquakes (e.g., Sabadini and Vermeersen, 1997).

Reconstructing past mantle states is fraught with substantial theoretical and practical challenges. In this study, we targeted some of these theoretical and practical hurdles by introducing G-ADOPT. Nevertheless, significant obstacles exist that are beyond the scope of this work. We predicated our work on zero uncertainty in our reference fields (i.e., the present-day tem-



perature field and past surface velocities), thereby committing what is known as the 'inverse crime' (Colton et al., 1998), a term used to describe the situation when the code employed in the inversions is also utilised to generate reference simulations.

Estimation of the present-day mantle state from seismic imaging and the assumptions regarding the thermal and compositional interpretation of seismic heterogeneity are both fraught with considerable uncertainty (e.g., Styles et al., 2011; Mosca et al., 2012; Zaroli et al., 2013; Davies et al., 2015). Furthermore, plate tectonic reconstructions can be uncertain, particularly further back in time and within the Pacific region (e.g., Shephard et al., 2012; Williams et al., 2015; Tetley et al., 2019). However, the existence of seafloor spreading isochrons up to approximately 125 Ma for all major plates provides confidence

in modelling relative plate movements in more recent geological periods (Seton et al., 2020). An uncertainty impact study carried out by Colli et al. (2018) posits that the presence of uncertainties causes reconstructed and reference flow histories to diverge exponentially back in time, with unrealistic structures materialising within and adjacent to thermal boundary layers. To minimise these impacts, Colli et al. (2018) advocate for terminating the optimisation after a few iterations. Here, however, inclusion of regularisation terms in the objective functional mitigates these impacts, effectively constraining the reconstruc-

tion to a smoother solution. This becomes particularly advantageous in real-Earth applications where observational constraints become sparser and more uncertain back in time. Without a smoothing term, the solution to the initial condition can contain high-frequency noise, which would diffuse over the course of the simulation. Smoothing therefore drives the solution towards a longer-wavelength initial state, whilst maintaining sensitivity to shorter wavelength information recorded in seismic tomography images. Moreover, by formally introducing past surface velocities into the objective functional, we infuse sensitivity

information that propagates further back in time, refining the flow trajectory, to improve the accuracy of reconstructions in the upper thermal boundary layer region. This sets our approach apart from the method used in previous adjoint reconstruction simulations (e.g., Vynnytska and Bunge, 2015; Zhou and Liu, 2017; Ghelichkhan et al., 2021), where the sequential-in-time nature of plate velocity assimilation can improve flow trajectories only forward in time.

Our study lays the foundations for exploring several unresolved geodynamical questions. Previous research has shown that

prescribing plate tectonic reconstruction velocities as a top boundary condition improves the precision of mantle reconstruction models and diminishes noise (e.g., Colli et al., 2015; Taiwo et al., 2023). Our framework formally incorporates these constraints through misfit terms, and future studies should compare this with other methods to find the most efficient way to integrate this valuable data. Earlier research advocates solving the reduced adjoint system, effectively considering velocities as insensitive to initial conditions (e.g., Ismail-Zadeh et al., 2004; Liu et al., 2008). The second-order Taylor remainder convergence tests,

examined herein, provides a robust foundation for evaluating the accuracy of such simplifications. Furthermore, our framework sets the stage for including hitherto unused observations within our inversions, such as geochemical constraints on mantle temperature and pressure (Ball et al., 2021). This stems from the design principle of *composable abstractions* in the software packages used in G-ADOPT, ensuring all components' modularity, interoperability, reusability, scalability, and maintainability. Specifically, Firedrake emphasises a clear separation between using the finite element method and implementing it. Dolfin-

Adjoint automates the derivation and computation of the adjoint systems using high-level symbolic language, ensuring the same advanced strategies that are applied for the forward calculation are utilised in the adjoints. Finally, ROL offers large-scale



optimisation algorithms that seamlessly integrates with Firedrake and Dolfin-Adjoint. Their integration through G-ADOPT is a groundbreaking development that opens-upadjoint problems to a new class of user and developer.

## 5   Conclusions

Reconstructing the spatial and temporal evolution of mantle flow is critical to understanding fundamental processes that depend on time-dependent interactions between Earth's surface and its deep interior. Progress with this endeavour requires the community to move from idealised forward models to data-driven simulations that rigorously account for observational constraints and their uncertainties, using an inverse approach. One way to achieve this, which has recently gained traction within the geodynamical modelling community, is through so-called adjoint methods, in which unknown model parameters can be

optimised to fit available observational data (e.g., Bunge et al., 2003). The development and validation of adjoint models for non-linear, time-dependent problems, however, is notoriously difficult (e.g., Gunzburger, 2000; Naumann, 2011).

Recent advances in computational sciences, centred around three novel software systems — Firedrake (e.g., Rathgeber et al., 2016; Davies et al., 2022), Dolfin-Adjoint (Farrell et al., 2013a; Mitusch et al., 2019), and the Rapid Optimisation Library, ROL (The ROL Project Team, 2022) — have allowed us to overcome this formidable practical challenge through

the Geoscientific Adjoint Optimisation Platform. G-ADOPT leverages and combines these automated systems to provide the optimised, scalable, efficient and portable adjoint-based research software infrastructure that will allow the community to tackle fundamental research questions via data-driven simulations.

Through demonstrated success in two sets of twin experiments, our study showcases the applicability of this approach for determining the unknown initial condition for mantle flow and simulating its subsequent evolution through space and

time. Our synthetic experiments formally exploit (through misfit terms) both present-day constraints extending throughout the computational domain (analogous, for example, to constraints on thermochemical structure provided by seismic tomography) and time-dependent constraints, albeit restricted to Earth's surface (analogous, for example, to plate tectonic reconstructions). We have also explored the impact of regularisation, which constrain measures of amplitude and complexity in the optimal solution. Taken together, such a combination provides a compelling way of deriving knowledge about our planet's past using

the power of distinct complementary datasets, that would not be possible by examining individual datasets in isolation. Despite the cases examined herein including some simplifications relative to real-Earth scenarios (e.g., ignoring compressibility and 3-D geometry), the design of G-ADOPT ensures that application to more realistic geometries or physical approximations can be achieved with only a few changes to the forward model in Firedrake (as already demonstrated by Davies et al., 2022), with the subsequent derivation and calculation of a fully-consistent adjoint automated through Dolfin-Adjoint. The second-order Taylor

remainder convergence test considered here strengthens the validation process, setting a new benchmark for future geodynamic adjoint frameworks, and enabling the rapid validation of adjoint models for more realistic scenarios.

While our study represents a notable advance, it does not eliminate all challenges associated with reconstructing past mantle states through the adjoint approach. Seismic tomography provides uncertain images of present-day mantle structure, due to limited resolving power (e.g., Davies and Bunge, 2006; Ritsema et al., 2007; Schuberth et al., 2009; Styles et al., 2011),





ambiguity in the amplitudes of seismic wave-speed anomalies (e.g., Zaroli et al., 2013), uncertainties in compositional and mineralogical models of the mantle (e.g., Connolly and Khan, 2016), and their non-unique interpretation in terms of temperature, composition or phase (e.g., Davies et al., 2012; Garnero et al., 2016; Richards et al., 2023): there is, therefore, substantial uncertainty when constraining the mantle's present-day temperature and density state using seismic observations. Additionally, the rheology of Earth's mantle remains poorly constrained. Laboratory studies on the stress-strain relationship of mantle rocks

are challenging, as laboratory strain rates are orders of magnitude greater than geologically relevant rates (e.g., Karato, 2010). Studies on glacial isostatic adjustment (GIA) (e.g., Mitrovica, 1996) and the geoid (e.g., Hager et al., 1985) offer complementary insights into the absolute value and depth-dependence of mantle viscosity, although constraints are non-unique (e.g., Colli et al., 2016). Observations of seismic anisotropy can shed light on the deformation mechanisms operating within the mantle, although this remains an area of active research (e.g., Nicolas and Christensen, 1987; Hedjazian et al., 2017; Eakin et al., 2023).

The G-ADOPT framework, however, allows us to transform these inherent challenges into opportunities: through the generation of observationally constrained digital twins of Earth's mantle, these parameters and their sensitivities can be evaluated. For example, by reconstructing the evolution of mantle flow in space and time, and comparing model predictions against geodetic and geological observations of epeirogenic motion (e.g., Friedrich et al., 2018; Ghelichkhan et al., 2018, 2020), constraints can be provided on mantle viscosity and density structure (e.g., Gurnis et al., 2000).

Traditional modelling frameworks in various Earth Science disciplines have often been tailored to specific governing equations, resulting in tools that lack portability and are confined to those disciplines. For instance, the development of adjoint-based schemes, pivotal in data-assimilation, sensitivity analysis, and design optimisation, has significantly advanced meteorology and oceanography, but have faced hurdles in other fields due to the challenges of derivation and implementation. While our study focused on geodynamics, the modular design of G-ADOPT embraces the principle of composable abstractions, a concept that

aims to transcend these barriers: The components within our framework are crafted to facilitate easy assembly and reuse, fostering methods and frameworks that are modular, interoperable, reusable, scalable, and maintainable. This design philosophy enables adaptation to diverse research fields with minimal programming alteration and adjustments to UFL expressions. Such flexibility ensures that technological advances can be swiftly and effectively translated across various geoscientific domains.

*Code and data availability.* For the specific components of G-ADOPT, including the full scripts of the simulations used in this paper see

https://doi.org/10.5281/zenodo.10050733 (Gibson et al., 2023). For the specific components of Firedrake project used in this paper see https://zenodo.org/records/10047031 (zenodo/Firedrake-20231027.0).

*Author contributions.* SG and DRD conceived this study, with all the authors having significant input on the design, development and validation of the examples and cases presented. All authors contributed towards writing the manuscript.



*Competing interests.* David Ham is chief executive editor of Geoscientific Model Development. The contact author has declared that neither they nor their co-authors have any other competing interest.

*Financial support.* This research has been supported by the Australian Research Data Commons (ARDC), AuScope, Geosciences Australia and the National Computational Infrastructure (NCI) under G-Adopt platform grant PL031. It was also supported by the Australian Research Council under grant nos. DP170100058 and DP220100173 and the Engineering and Physical Sciences Research Council [grant no. EP/R029423/1]

*Acknowledgements.* Acknowledgements. Numerical simulations were undertaken at the NCI National Facility in Canberra, Australia, which is supported by the Australian Commonwealth Government. The authors are grateful to the entire Firedrake, Dolfin-Adjoint and ROL development teams for support and advice at various points of this research.



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



**Figure 6.** Iterative optimisation process visualised: A and B depict the reference initial and final conditions. C-F present the reconstructed initial conditions at the 0th, 20th, 50th, and 100th iterations. G-J highlight the misfits, representing the squared differences between the reconstructed initial temperature fields and the reference temperature. Similarly, K-N display the reconstructed final temperature fields after the 0th, 20th, 50th, and 100th iterations, with their respective misfits demonstrated in O-R.



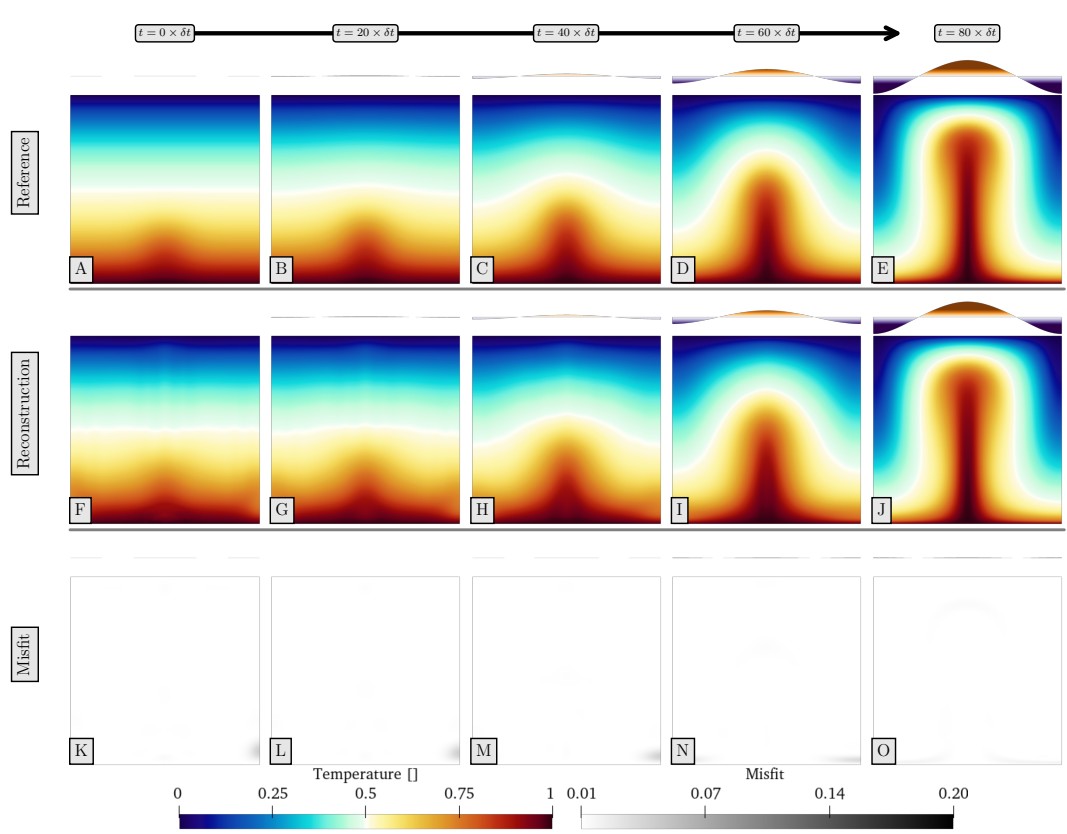

**Figure 7.** Comparison of reference forward and reconstructed simulations over time: A-E present the evolution in the reference forward, while F-J depict the evolution in the reconstructed simulations. The misfits between the reference and reconstructed scenarios at each time step are illustrated in K-O. Note that surface dynamic topography is represented by visualising the normal stresses at the top boundary.





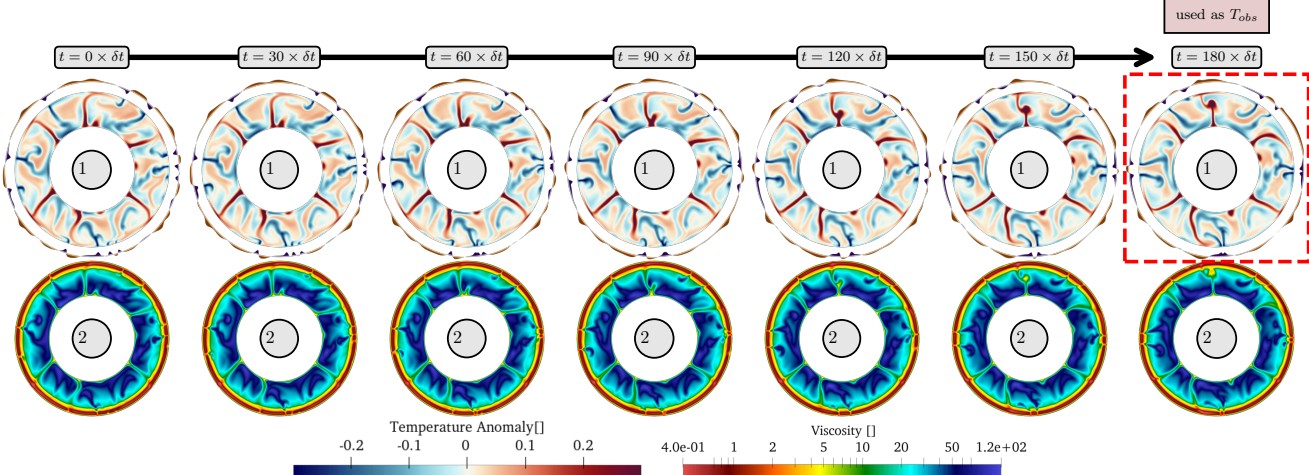

**Figure 8.** Reference forward simulation spanning a duration of $t = 200 \times (5 \times 10^{-6})$, as depicted in 1.A to 1.G. Figs. 1 (upper panel) detail the temporal evolution of the reference temperature field, while Figs. 2 (lower panel) show the viscosity field at each time. The viscosity demonstrates a variation of nearly three orders of magnitude: approximately $1.0$ in the asthenosphere, $140$ within colder slabs of the lower mantle, and $0.4$ in regions exhibiting elevated principal strain-rates within the convergent zones.

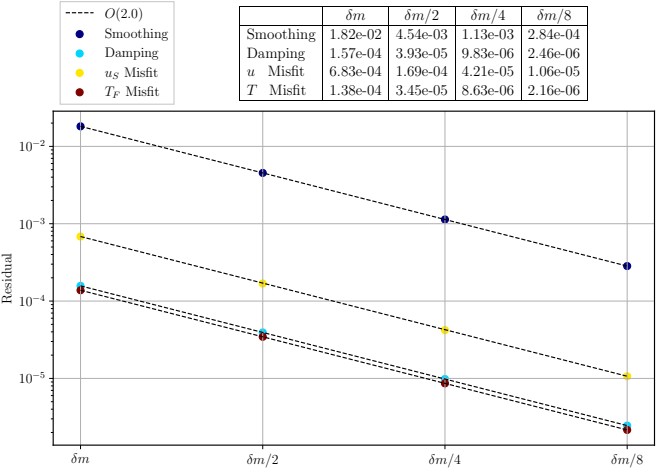

**Figure 9.** Second-order Taylor remainder test for convection with temperature, stress-dependent rheology.





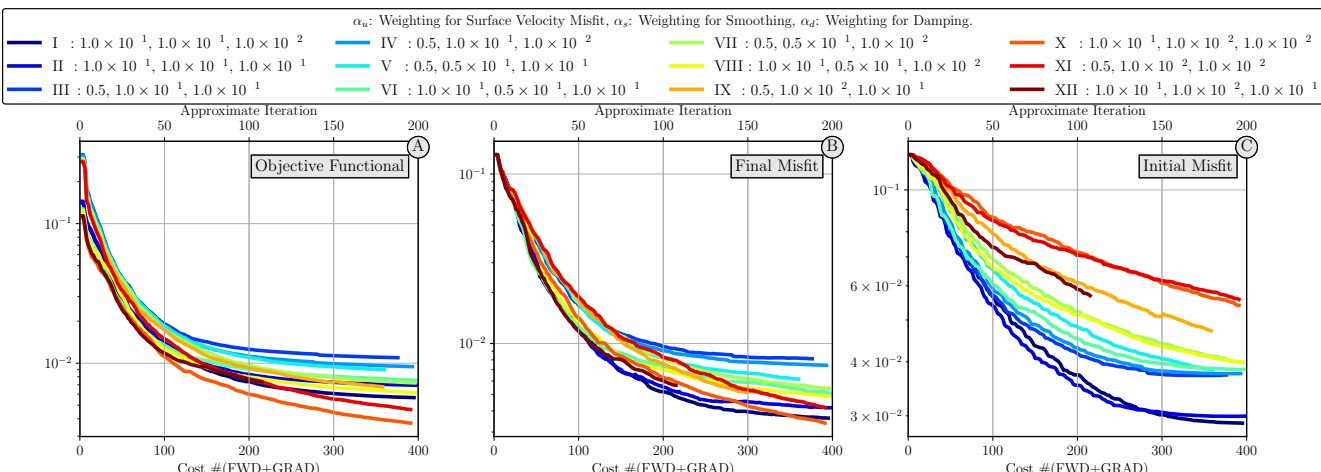

**Figure 10.** Summary of optimisation outcomes from Earth-like experiments. (A) visualises the process of objective functional minimisation. (B) illustrates the final misfit, representing the misfit between the reconstructed final temperature field and $T_{obs}$. (C) depicts the initial misfit, indicating the difference between the reconstructed initial condition $T_{IC}$ and the reference initial condition. Notably, despite significant reduction in the objective functional and final misfit in simulations XI and XII, these simulations do not perform as well in terms of the initial misfit, which is a key measure in our experiments.





**Figure 11.** Comparing the reference (1 and 2) and the best reconstruction simulation (3 and 4). Temperature and viscosity fields are shown in panels 1,3 and 2,4 respectively. The misfit, which is the squared difference between the reconstructed and reference temperatures, is shown in panel 5. To highlight the effectiveness of the reconstruction of the evolution of surface dynamic topography, a field representing the normal stresses acting on the top boundary is visualised alongside the temperature fields in 1 and 3. The reconstruction simulation employs values of $\alpha_u = 10^{-1}$, $\alpha_s = 10^{-1}$ and $\alpha_d = 10^{-2}$, and is marked with I in Fig. 10.