# Peer review of "Automatic adjoint-based inversion schemes for geodynamics: Reconstructing the evolution of Earth's mantle in space and time"

_EGUsphere, 2023_

## Community Comment (CC1)

**Review of « Automatic adjoint-based inversion schemes for geodynamics: Reconstructing the evolution of Earth's mantle in space and time » by Ghelichkhan et al.**

The manuscript presents a workflow to realize a geodynamic grail: full inversion of mantle flow together with tectonics to generate digital twins of the Earth. The paradigm in the past to was to push plates at the surface of rheologically simple models and either push them as long as possible so that the last tens of Myrs could become adequate or employ and adjoint for an inversion of the final state given by a thermo-chemical translation of seismic tomography images. Inverting surface kinematics was proposed recently by Li et al. (2007) and Bocher et al. (2016, 2018) on similar types of models (2D). What does this manuscript bring compared to the former studies? It proposes an adaptive framework thanks to automatic differentiation and open software machinery. One could argue for the adjoint method keep physical consistency over the reconstructed duration. The generalization of the approach to other geodynamic problems makes it also very interesting for the community. I find the paper is well presented, provides most of what is needed and giving the most of the needs to evaluate if one wants to dive into the inverse geodynamic adventure.

I have general recommendations for improvements.
 - The most important one is how the Taylor test (or gradient test) is performed. Indeed the residual has to be O(h^2) WHEN h tends to 0. So one can find that it is O(h^2) for some values of h, but it is the most important that it is O(h^2) at the vicinity of machine epsilon. Olivier Talagrand who pioneered the adjoint methods for data assimilation was hard on me on this issue and sent me a variety of gradient test that are successful and failing. Some of them show a residual O(h^2) for large h and become O(h) close to h. So the authors should diminish delta m by orders of magnitude until they reach machine epsilon and the residual becomes then stable. If the residual is O(h^2) close to when it is close to machine epsilon, everything is good. Many papers show this kind of plot. For geodynamics, you can check my preprint on automatic differentiation of StagYY (https://eartharxiv.org/repository/view/6398/ - it can be also interesting to include it in your discussion since we both took parallel paths). By doing so, Figures 4 and 9 will leave no doubt on the efficiency of your method. I you want me to send you the document of Olivier, let me know.
 - The automatic differentiation section (Dolfin). This is very important to be more explicit since it is a new methodology that you take to geodynamics, and this is precisely this methodology that empower your workflow. It is too abstract now. I would prefer you explain more precisely how it works and how it respects the idea that the generated adjoint is the exact adjoint of the forward code. I understand it is hard, since it is why I have such hard time publishing my preprint. But a little more would be nice. Maybe I did not have enough time, but it seems to me that some of the important preparation work to make for automatic differentiation is not explicit. Do you have directives in the forward code? Where do you tape, or checkpoint? What is your organization? It is nice to give a glimpse to the future users of what to be careful about if they want to modify what you have done? Off course, if it's too technical in the paper, that may not work.
 - I would put the derivation of the adjoint equation in an Appendix, you could refer to when you talk about the linear properties of the adjoint with pseudo-plasticity.
 - In my opinion, the discussion needs reworking. For now, most of it reads like a conclusion. The results should be compared to former studies. Bocher et al. (2016, 2018) are ideal because the models have similarities in geometry and objectives. You can discuss then, how the inversion performs. You have all the results for this in your figures. You have to discuss also computing time. This is very important also because you state you want your method to tackle grand challenges of 3D spherical reconstruction with self-generating plates. For now your forward model is 2D, with only 3 orders of magnitude variations in viscosity, which is not enough yet to have plate-like behavior. What is the computing power needed for inversion of this 2D case? To move on, how do you envision going to 6 or more orders of magnitude and 3D? How many million years will you be able to invert for given your computing time? This is a necessary discussion here, since you bring the ambition of the method from the start.

Recommendations along the text:
- Davies and Richards, 1992 in the intro: you should propose a more up to date review here, since so many discoveries have been made since.

- l.36: « state is unknown »: not only. Also because the physics we use is incomplete, and the physical and chemical properties of the Earth remain uncertain.
- l.39: Rolf and Tackley citation here does not feel appropriate. If you want to add a citation with pseudo-plasticity, you souhld choose one focusing on the equations.
- l.43: all the citations here are from the same group, except Dannberg, although other groups have used state-of-the-art 3D convection calculations at high resolution with different codes. Actually, you cite some of them in the next paragraph so you could cite them here too.
- l.44: I would remove the discussion on Monte-Carlo technique. It's not necessary here. It feels out of the scope.
- l.50: Bocher et al. (2016) is a Kalman filter. So it should be cited later. Bocher et al. (2018) is ensemble Kalman filter. Both these methods are doing a similar work as you propose later so they are useful for you to compare. The major difference is that we did not use exactly the same « observations » for the inversion and did apply a Kalman smoother to correct past states, which is something you are interested in.
- l.53. « assimilated » is not a correct word here. I know it has been misused for years. But now you are considering REAL assimilation of surface kinematics, you see the problem. What you call assimilation here is forcing the surface of a model, so it is a forward operation. While assimilation in data assimilation is an inverse operation, specifically the one you are designing here. So please, change it to « nudged » or some more appropriate term.
- Figure 1. In line with the « assimilation » word. Case (a) can be a « nudged » model. It is a way to differentiate nudging (a), KF data assimilation (b) and 4Dvar data assimilation (c).
- L.62: Kalman smoother is a Kalman filter method to do retro-propagate the information in past states. So it is possible to use Kalman filter and improve the knowledge on initial conditions. The major difference with the adjoint is that it is a statistical treatment whereas the adjoint keeps the physical consistency (it has its advantages and disadvantages, considering that the physical model is not perfect).
- l.65: I would remove that idea of an emerging field. We could say that the inverse models of the geoid back in 1990s were inverse geodynamics. Or the pionnering works of Peter and Alik are 20 years old already and we cannot say the technique has penetrated the geodynamics community. Actually, what you propose could help to do so.
- l.74. About Li et al. (2017). It can be important to be nuanced saying that such recovery is limited by the ill-posedness, which is their conclusion.
- Intro: I would say in the intro that solving adjoint equations does not necessarily generate the exact adjoint of the forward code, while AD does. Hence, your approach enforces that the adjoint code is the exact adjoint of the forward code, which is necessary for optimal inversion.
- Table 1: Delta T = Temperature difference between top and bottom ; d= thickness of the system. It is not just characteristic something.
- l.219: it would be nice to cite Talagrand, 1997 or some of his pioneering work here.
- l.225-230: Okay for the advantages. But which drawbacks? --> not the exact adjoint of the forward code which may not be good for inversions. Write a new code for each new problem (set of equations).
- l.231: I think here you could start with AD right away because with AD you find a solution to both the limitations of the continuous adjoint: exact adjoint and versatily of problems (not rewriting the code if you change the problem/update the code). I would reduce the length and you go straight to the point.
- l.233: you can cite Giering and Kaminski, 1998 for AD pioneering work.
- l.240: too bad you forget to cite some existing AD tools used for science application purposes: Tapenade, TAF, or Enzyme.
- l. 240 and equivalence of the methods: I don't agree: the solvers, the methods involve a suite of operations that are discrete anyways and can involve some differences that can possibly prevent the continuous adjoint to be the exact adjoint. It's not only a question of resolution but the approximations you make also, and the tools you use to solve the problem.
- 2.2.2 -> More would be needed here. It's a major point of your work.
- l.560: I find your forward model to be a little oversold here. With 3 orders of magnitude viscosity change, you don't make plates in convection models, and it's even more unlikely to have Earth-like subduction zones (one-sided etc…). Using this model is perfect for the demo here, but I am sure you want to have a more plate-like behavior in the mode in the future, which means necessarily increasing this contrast.
- For the duration of your model, it is worth giving a glimpse of what Earth time it corresponds to, so that we can evaluate how far back in time we can go given the uncertainty on the past state

we look for. Simply putting dimensions here it would be close to 200Myrs. But probably the convection velocities are lower when I look at your figure and it would probably translate to half of this maybe? That would be useful.

- l.612: Visco-plasticity means surface decoupling and more non-linearities. It matters and you could show it by showing the adjoint: the sensitivity of your misfit to surface velocities is probably lower than the isoviscous case. But again, 3 orders of magnitude is not really plate-like so pushing the interpretation is not necessarily good here.
- l.563: surface velocities, except today, are not observed. They come from kinematic models.
- 708: Seton et al., 2023 provides an overview of plate reconstructions that should be discussed here.
- 711: It's not an inversion, but the work of Bello et al. (2015) assesses in some ways the errors produced on the forward modelling.
- 715: The issue with smooth solution is that geodynamic features are sharp and their shape is fundamental for the self-consistency of the flow, especially with variable viscosity. This is very different from seismic or gravity inversions in which smoothing does not hinders the fundamentals of the physics at play.
- l.723: not « assimilation » what you do now with surface velocities is real assimilation :)
- Conclusion can be reduced to the essentials.

Nicolas Coltice

---

## Author Comment (AC1)

Responses to Reviewer Comments for:

**Automatic adjoint-based inversion schemes for geodynamics: Reconstructing the evolution of Earth's mantle in space and time**

Sia Ghelichkhan, Angus Gibson, D. Rhodri Davies, Stephan C. Kramer, and David A. Ham

March 30, 2024

We would like to express our sincere gratitude to the three reviewers, Georg Reuber, Nicolas Coltice, and an anonymous reviewer, for their insightful and constructive comments. Their expertise and thoughtful suggestions have allowed us to enhance the quality and clarity of our manuscript. We also extend our thanks to the journal's editorial staff, specifically Boris Kaus, for his efficient handling of the review process and the understanding and flexibility regarding circumstances beyond our control. Below are our responses marked with "Response - [#] ", each following the original comment it is addressing in light italic grey.

**Responses to Reviewer 1: (Georg Reuber)**

G. R. – C1: *" The authors present an open-source mantle convection inversion modelling library based on Firedrake, ROL and dolfin-adjoint. The library makes use of Firedrake's high level of abstraction to discretize the forward problem and automatically derive the adjoint equation and derivative expressions via UFL. A connection to ROL allows for usage of scipy optimizers. They solve 2 toy inverse problems while investigating the effects of their regularization choice. The contribution is valuable to the community as their library might facilitate prototyping of mantle convection related inverse problems. The choice of regularization terms can be adapted for future (mantle) reconstructions. The paper is very well written and mostly well structured. Especially the description of the numerical implications of DTO versus OTD is well formulated. I recommend publication with minor modifications. "*

Response – 1: We appreciate Georg's positive feedback and concise summary of our work. To add to this, we would like to reiterate a detail that has been appropriately addressed already in our manuscript. The Rapid Optimization Library (ROL), as described in our manuscript, is a component of the Trilinos project, designed for highly-efficient, large-scale optimisation, and provides an alternative rather than a interface to scipy's optimization algorithms. Although many of ROL's algorithms are similar to those in `scipy.optimize`, there are fundamental differences. Most importantly, ROL provides a generic interface for data structures that can be overloaded to perform inner-product aware operations (e.g., in L2 space) and achieve mesh-independent convergence results not otherwise attainable with `scipy.optimize`.

G. R. – C2: *" Major comments: Collect all mathematical symbols in Table 1."*
Response – 2: In response, we have updated Table 1 to include all mathematical symbols up to Section 2.2.3. We opted not to incorporate the symbols in the adjoint system's derivation into Table 1 to maintain readability and prevent it from becoming overly extensive.

G. R. – C3: *" A comparison to existing automatic adjoint frameworks for PDEs, e.g. FEniCSx adjoint, TAO based PETSc adjoint, Julia AdFEM, etc might be valuable. On one hand it remains unclear what parts the library are abstracted away from a potential user in contrast to directly using Firedrake, FEniCS, PETSc etc. On the other hand, it is unclear what the boundaries on the flexibility of the library are. Such a discussion on the library would facilitate potential users' choice to use the author's library for their inverse problem at hand."*
Response – 3: Acknowledging the reviewer's suggestion for a comparison with existing AD frameworks for PDEs, we note the variety of AD tools that have been developed over recent years. Tools such as FEniCS utilise Dolfin-Adjoint for AD, similar to the approach employed by Firedrake. The manner in which solve options and adjoint systems are managed, however, differs between Firedrake and the broader FEniCS projects. As for the specific FEniCSx project,

it is a new implementation of the FEniCS project that is currently under development. As of the last public update, there is currently no automatic differentiation functionality available in FEniCSx. Finally, any detailed description of Firedrake and other FEniCS projects is outside this paper's scope and is discussed in our previous work (Davies et al., 2022).

Regarding the Toolkit for Advanced Optimization (TAO) within PETSc, the focus is again on providing algorithms for solving large-scale optimisation problems in high-performance computing contexts, similar to ROL. Similarly, TAO is geared towards utilising gradient information supplied by the user or computed via other methods, such as finite differences or AD tools, rather than directly implementing AD functionalities. Automatic differentiation is indeed one of the major focus areas identified by PETSc's road-map, which is achieved by developments such as PETSc TSAdjoint, which is a high level differentiation processes, similar to pyadjoint.

JuliaAdFEM represents another sophisticated computational framework designed for solving PDEs, leveraging Julia's just-in-time compilation and the multiple Julia's built-in AD tools.

A detailed comparison of these frameworks would necessitate a thorough explanation of their distinct principles, which we feel goes beyond the scope of this manuscript. Our discussion on AD aims to convey how it enables accurate derivative calculation while maintaining performance *a core objective of our platform designed to abstract such complexities from the geodynamic end-user.*

G. R. – C4: *"Some questions that remained unclear to me are: How to include and load data - is it limited to the data that the authors present toy inversions for? Does changing any rheology require function overloading or is the weak form and constitutive laws an input? What types of BCs are pre-implemented? Are other time discretizations pre-implemented or does a potential user have to provide it? Particle advection included and in what form? How to configure the actual solver (multigrid etc?) Visualization capabilities Support for complicated initial meshes, e.g. via gmesh interface. Even if these questions are answered in previous work of the authors, it would be helpful to mention included functionality and potentially cite the work again."*

Response – 4: We thank the Reviewer for highlighting the need for more detail on our software's general features. While the combination of online documentation both provided by Firedrake and G-ADOPT projects and the paper by Davies et al. (2022) provide extensive information, we acknowledge the importance of summarising key functionalities in our manuscript. Here's a brief overview, with more details available in our online documentation for G-ADOPT and Firedrake:

- **Data Inclusion and Loading:** This is facilitated by Firedrake's built-in features, as detailed in the Firedrake documentation here.

- **Rheology and Weak Form Inputs:** By employing UFL for describing rheology and constitutive equations, G-ADOPT automatically adapts to any rheological law used in the forward model, ensuring consistency in the adjoint system.

- **Time Discretization Algorithms:** G-ADOPT offers various schemes, including second-order Runge-Kutta, backward/forward Euler, and implicit midpoint. Users can also implement custom algorithms.

- **Particle Advection:** This is an ongoing development in Firedrake, and future enhancements will be integrated into our platform. Current developments are focused on Finite-element consistent treatment of points cloud in Firedrake known as *VertexOnlyMesh* in Firedrake, and their movement(Nixon-Hill et al., 2023).

- **Initial Meshes and Visualisation:** Firedrake supports complex meshes (e.g., via gmsh) and visualises outputs through seamless integration with *Visualization Toolkit* (VTK), with limited inline visualisation capabilities through Matplotlib.

- **Boundary Conditions:** G-ADOPT supports the commonly adopted mantle flow problem boundary conditions, i.e. free-slip and no-slip on both curved and Cartesian boundaries, together with innovative boundary conditions such as a free surface. The details for the former are described in our previous paper (Davies et al., 2022), while the latter is the focus of on-going developments within G-ADOPT.

G. R. – C5: *" Also, as the code is open source, the code screenshots as well as their detailed description could then be moved to an Appendix."*

Response – 5: We appreciate the reviewer's suggestion and acknowledge that moving the code listings to an appendix

could improve readability. However, our decision to include code listings in the main text was informed by GMD's guidelines. These guidelines state that "all material required to understand the essential aspects of the paper, such as experimental methods, data, and interpretation, should preferably be included in the main text". Given their minimal nature, these listings are pivotal in illustrating the use of our platform, particularly in aspects like the selection of finite element function spaces and time-stepping schemes. They also allows us to concisely demonstrate the ease of use, which is one of the main motivations for our developments.

G. R. – C6: *" 447 & equation 8. As you point out, this will also act as a regularizer on the reconstruction. Can you elaborate what (types of) reconstructions this will affect?"*
Response – 6: Our rationale is grounded in the principles of Total Variation Regularisation (TVR), a regularisation technique widely used for its de-noising capabilities. Such regularisation approaches penalise the aggregate variation within the solution, thereby concentrating on the solution's total variations. By restricting our control to the $Q_1$ space, we achieve this objective. Analogous to TVR, this strategy results in de-noising of the solution by mitigating higher variations associated with higher polynomial degree basis functions, in the $Q_2$ case.

G. R. – C7: *" The analysis on finding the optimal scalings for the regularization, as well as velocity term respectively, is interesting (even though generalization of it might be limited). As the authors put some focus on the effect of the regularization, visualizing the reconstructions of their scaling parameter search to show the effect of the regularization can be interesting. This could also be done in an Appendix."*
Response – 7: We agree with the reviewer on the significance of our findings regarding optimal scaling for regularisation and the velocity term, recognising the limitations in generalising these results. We believe a more comprehensive parameter search involving factors like Rayleigh number, temperature and strain-rate rheology dependencies, and simulation duration, would be necessary to fully generalise our findings. However, undertaking such an extensive search and including the results in an Appendix would not align with the guidelines provided by GMD, making it impractical for this publication. This presents an opportunity for future research, where a detailed exploration of these parameters will be pursued.

G. R. – C8: *" Even though the code is open source and potential users could do performance tests themselves it would greatly facilitate the choice to use the author's library if they would provide runtime performance results, as well as more general information, like:*

1. *Wall clock time of their simulations*

2. *Used hardware*

3. *Integration points per element*

4. *Parallelization capabilities (e.g. matrix free solvers for forward Newton and inverse Newton, full PETSc MPI support?)*

5. *Functionality of FEniCSx and PETSc incorporated and abstracted to what level? (e.g. solver availability? E.g. all multigrid solvers from PETSc configurable, if geometric multigrid how to discretize etc?)*

*"*

Response – 8: We thank the Reviewer for their inquiry into technical aspects. We acknowledge the importance of providing performance metrics to aid users in evaluating our library. Our platform undergoes regular runtime performance tests as part of our repository's *GitHub Actions*, measuring the runtime of various case studies from this paper and the previously published work by Davies et al. (2022). These tests are conducted on a dedicated cluster unit and at the National Computing Infrastructure in Australia. The results are readily accessible on our repository, for example by clicking here. Below are detailed responses to the specific queries raised:

1. The runtime for our documented use cases is consistently evaluated and updated on our GitHub page. For precise runtime information, users can refer to the efficiency ratios detailed in our publication (representing the maximum theoretical values) and compare them with the runtime metrics available for each case on our GitHub Actions, including simulations that mimic real Earth scenarios.

2. The simulations were performed on *Gadi*, a supercomputer at the Australian National Computational Infrastructure (NCI), utilising Intel Sapphire Rapids processors.

3. Routine regression tests via our GitHub Actions are executed on a dedicated cluster powered by an *AMD EPYC 7763 64-Core Processor*. The manuscript's simulations, however, are conducted on Intel Sapphire Rapids, demonstrating improved runtimes to those available online.

4. Regarding parallelisation capabilities: G-ADOPT leverages Firedrake and its integration with Dolfin-Adjoint. Firedrake itself utilises the distributed memory parallelism withing PETSc. Through coupling with petsc4py, it seamlessly integrates with PETSc, enabling strategies like matrix-free solvers. These functionalities are elaborated in our previous publication by Davies et al. (2022) and other references for Firedrake. We direct readers to these sources for comprehensive details.

G. R. – C9:*" The data types used in the toy inversions are explained in the discussion section. To facilitate reading one could move this part to a separate section, potentially before presenting the inversions. I agree with the authors that a substantial discussion of the used data types is beyond the scope of this work, which should mainly present their library. Nevertheless, a discussion of availability and quality of (current) global (on each grid point) temperature data might be helpful, as it might be strongly priored by a previous model."*

Response – 9: We concur with the reviewer on the significance of discussing the datasets utilised for the inverse methods outlined in our manuscript. Considerable thought has been devoted to how best to integrate this suggestion, including the possibility of segregating the discussion concerning data. Nevertheless, the exposition of data within the discussion section primarily serves to underpin some of our choices in the inverse scheme, notably the selection of regularisation terms. The body of recent research, to which we have tried to duly cite throughout the manuscript, extensively examines these datasets. Therefore, we posit that retaining the current structure of the discussion serves best to preserve the link between the datasets and our methodological decisions.

G. R. – C10:*" Minor comments:*

- *Note that there is no guarantee that automatic differentiation, even on the symbolic level, leads to the maximum possible efficiency, e.g. derivations by hand might, in complex systems, still find more efficient substitutions.*

- *Equation 7: 7a $u_{obs}$ should be a vector (bold). Also, squaring $u_{obs}$ probably means taking the inner product here?*

- *Explain further your choice to normalize against equation 7a, or temperature, and not using a standard scaling for all terms*

- *91: Sentence is hard to understand.*

- *168: ther – their*

- *430: should the velocity vector u be bold?*

- *Figure 3 is missing letters in the subfigures*

- *Figure 6: Missing characters in the text boxes on the left.*

- *Typo: Inital − > Initial*

- *738: Missing space*

*"*

Response – 10: What follows are our responses: In response to the Reviewer's minor comments, we provide the following clarifications and corrections:

- We concur with the reviewer's point regarding automatic differentiation. As discussed in Section 2.2.1, manually deriving adjoint equations can indeed lead to more cost-effective approximations. However, our derivation of the discrete adjoint system suggests the runtime of one forward simulation with a linearized timestepping scheme as the upper efficiency bound, primarily due to the similarity of the adjoint and forward systems. Manual approximations, while potentially exceeding this theoretical limit by introducing some approximations, might compromise the accuracy of gradient information.

- Correction made. The squaring of $u_{obs}$ intended to denote the inner product $u \cdot u$ has been appropriately revised.

- The rationale behind normalising against the first term (associated with final temperature field) stems from prior studies that dealt solely with this term. In those studies, given that the control (initial temperature field) shares the same units as the state variable temperature, no normalisation was deemed necessary. We normalise all subsequent terms relative to the first to ensure comparability.

- The sentence at line 91 has been rephrased for improved clarity.

- Typographical error corrected.

- The presentation issues in Figures 3 and 6 were attributed to the use of PDF format for figures within the LaTeX compilation. We have transitioned to PNG format to address this.

**Responses to Reviewer 2: (Anonymous)**

Anon–C1:  *I enjoyed reviewing this manuscript, which explores the performance of adjoint mantle convection models aimed at initial condition recovery in the context of a powerful new computational modeling framework, i.e. the Geoscientific Adjoint Optimisation Platform (G–ADOPT). The MS is well written, the figures are effective, the results are well presented and should be of broad interest to the geodynamic modeling community. To say it upfront: I strongly urge publication as is. Three advances make the MS particularly noteworthy.*

- *The authors introduce the Geoscientific Adjoint Optimisation Platform (G- ADOPT) as an efficient computational modelling platform. They also present a number of impressive computational and accuracy performance measures. This will be helpful as a wider geodynamic user community wishes to adopt adjoint based methods for their work.*

- *The authors introduce the effects of damping and smoothing to the inverse problem.*

- *The authors introduce a surface velocity misfit into the objective functional. The spatial extend of the associated kernel differs from the thermal misfit. Importantly, the measure injects misfit information throughout the simulation period. This makes the new measure complementary to the mere use of the final thermal misfit information applied in earlier adjoint models. I regard this as an important step forward. In my copy of the MS, the labels of Figure3 A−D are not printed, this needs to be fixed*

Response – 1:  We are grateful for the reviewer's insightful feedback and appreciate the positive summary of our work. Regarding the issue of missing labels from Figure 3 A–D, this was indeed due to the use of PDF format for figure compilation. We have since transitioned to PNG format to address this issue. Below we address the specific points raised in the review.

Anon–C2:  *I have three minor comments: 1) The lower misfit reduction in the non-linear experiment relative to the isoviscous experiment is attributed both to the higher Rayleigh number and to the extended simulation period. The latter appears to approach a transit time, although this seemingly is not made explicit. However, a higher Rayleigh number should help in the adjoint performance, as unrecoverable effects from thermal diffusion are reduced. So it would seem that the longer simulation period is the more relevant factor when explaining the lower misfit reduction.*

Response – 2:  We agree with the reviewer's observations regarding the diminished misfit reduction in our non-linear experiment. In response, we now focus on the extended simulation time as the key factor in our discussion of results.

It is important to note, however, that while high Rayleigh number fluids exhibit reduced diffusion effects – thereby minimising information loss – they are also characterised by increased chaotic behaviour, as highlighted in previous research (Bello et al., 2014). This chaos, although mitigated in synthetic adjoint inversion tests through the imposition of known surface velocities (Colli et al., 2015; Taiwo et al., 2023), does not necessarily improve initial condition recovery. Our study's new approach, where surface velocity emerges from the inversion process itself, leaves the chaotic nature of high Rayleigh number simulations unaltered, adversely impacting the results over prolonged simulations.

Future efforts will aim at strategising the incorporation of various information sources, such as initially imposing surface velocities and progressively augmenting the objective functional in later iterations to enhance solution recovery and consistency.

Anon–C3:  *2) Surface velocity misfits are helpful for initial condition recovery if the surface velocities exclusively represent the influence of mantle flow. There are observations (e.g. Late Miocene slow down of South American plate velocity) that this may not*

*be the case.*

Response – 3: We acknowledge the reviewer's insightful comments regarding surface velocity misfits. It is understood that the surface velocities observed on Earth result from the combined forces arising from thermal and chemical anomalies across the mantle, and tectonic forces acting at plate boundaries, including orogeny at convergent boundaries. Previous studies have indeed associated the rate of change in observed surface velocities – not the absolute velocities assimilated here – with variations in plate boundary forces (e.g., slab sinking, orogeny) or changes in asthenospheric pressure-driven flow. Our model does not incorporate the forces related to plate geometries, such as orogenies, which means surface velocities driven by these boundary forces could be misattributed to other forces present in our models.

In light of this, we propose two essential considerations regarding surface velocities:

1. Surface velocities should be considered as a secondary dataset in data assimilation methods, supplementary to the primary temperature field derived from seismic tomography. While plate reconstruction models offering surface velocity reconstructions provide a valuable perspective on past mantle flow, their surface-limited nature means they serve primarily as a proxy for the aggregate force balance in the mantle.

2. The dynamic forces influencing plate movements, which may not be accounted for in our models, highlight the necessity for employing appropriate weighting factors. These factors should prioritise direct temperature observations from seismology (i.e. final temperature field) over the derived observations from plate reconstruction models, despite the latter being crucial for understanding geologic force balances.

To clarify this point, we have referenced studies by Colli et al. (2014) and Iaffaldano and Bunge (2015) in our discussion and have expanded the manuscript's discussion section accordingly. This approach acknowledges the limitations and challenges in interpreting surface velocity data and underscores the importance of integrating direct observations with model-derived insights to more accurately infer mantle dynamics.

Anon–C4: *3) The authors rightfully acknowledge the inverse crime, i.e. their experiments neglect effects from uncertainty in the use of the forward model and the relevant observations. The inverse crime effects in geodynamic adjoint models where studied explicitly in Colli etal. 2020 and that study could be cited.*

Response – 4: The reference to the study by Colli et al. (2020), which explicitly investigates the effects of inverse crimes, was indeed intended to be a crucial part of our discussion on the complexities of time-dependent reconstruction models in geodynamics. Due to an oversight on our part, the manuscript originally referenced Colli et al. (2018), instead of the correct study from 2020 in the discussion between lines 706 and 710. We apologise for this confusion and have since corrected this mistake in our manuscript to accurately reflect the significant contributions of Colli et al. (2020).

**Responses to Reviewer 3: (Nicolas Coltice)**

N. C.–C1: *The manuscript presents a workflow to realize a geodynamic grail: full inversion of mantle flow together with tectonics to generate digital twins of the Earth. The paradigm in the past to was to push plates at the surface of rheologically simple models and either push them as long as possible so that the last tens of Myrs could become adequate or employ and adjoint for an inversion of the final state given by a thermo-chemical translation of seismic tomography images. Inverting surface kinematics was proposed recently by Li et al. (2007) and Bocher et al. (2016, 2018) on similar types of models (2D). What does this manuscript bring compared to the former studies? It proposes an adaptive framework thanks to automatic differentiation and open software machinery. One could argue for the adjoint method keep physical consistency over the reconstructed duration. The generalization of the approach to other geodynamic problems makes it also very interesting for the community. I find the paper is well presented, provides most of what is needed and giving the most of the needs to evaluate if one wants to dive into the inverse geodynamic adventure. I have general recommendations for improvements.*

Response – 1: We are greatly encouraged by the Nicolas' comments and thank him for the insightful summary. We share the enthusiasm that our open-source framework will serve the geodynamic community well, facilitating the exploration of a broad spectrum of geodynamic problems.

N. C.–C2: *The most important one is how the Taylor test (or gradient test) is performed. Indeed the residual has to be $O(h^2)$ WHEN h tends to 0. So one can find that it is $O(h^2)$ for some values of h, but it is the most important that it is $O(h^2)$ at the vicinity of machine epsilon. Olivier Talagrand who pioneered the adjoint methods for data assimilation was hard on me on this issue and sent me a variety of gradient test that are successful and failing. Some of them show a residual $O(h^2)$ for large h and become $O(h)$ close to h. So the authors should diminish delta m by orders of magnitude until they reach machine epsilon and*

*the residual becomes then stable. If the residual is $O(h^2)$ close to when it is close to machine epsilon, everything is good. Many papers show this kind of plot. For geodynamics, you can check my preprint on automatic differentiation of StagYY Link - it can be also interesting to include it in your discussion since we both took parallel paths. By doing so, Figures 4 and 9 will leave no doubt on the efficiency of your method. I you want me to send you the document of Olivier, let me know.*

Response – 2: We appreciate the reviewer's input on conducting Taylor Remainder tests and their referral to their preprint, which we now use as a reference in our discussion. We are also grateful for the shared experiences and insights from Olivier Talagrand, a pioneer in variational data assimilation within Earth sciences.

Addressing the concerns raised, we've extended the perturbation range in our Taylor Remainder tests to $10^{-10}$. The results demonstrate the robustness of our approach, ensuring that our analysis comprehensively extends down to smallest floatig-point number, here defined as smallest positive $\epsilon$, such that $1.0 + \epsilon$ is differentiable from $1.0$. As depicted in Fig. 1, both the top and bottom outcomes reveal deviations from the expected $O(h^2)$ behaviour as residuals approach $\epsilon$, a behaviour already documented in Coltice et al. (2023). Notably, in cases where we use SNES to solve non-linear stokes problems, when setting the relative solver tolerance to $10^{-5}$ (shown by transparent yellow and red markers), perturbations beyond solver tolerance exhibit divergence from $O(h^2)$ trend, primarily dictated by the precision of our forward system solutions. When setting this tolerance to $10^{-10}$ we see that the same behaviour as reported by Coltice et al. (2023) is observed. Using direct solver options, i.e. MUMPS, sets the precision limit in the isoviscous case to practically machine precision.

These tests underscore a crucial consideration in applying adjoint methods within geodynamics: the practicality and precision of derivative calculations are significantly influenced by the solver tolerances applied in both forward and adjoint processes. Although achieving machine-level precision in derivative fields remains ideal, the complexity of real-world simulations often renders this goal unattainable due to computational constraints. Our findings highlight this reality, underscoring that while our method is capable of maximum accuracy, the limitations imposed by solver tolerances are an inescapable aspect of adjoint inversions, which question the necessity of approaching machine precision accuracy for the derivative information. Nonetheless, the results presented are aligned with expectations from Nicolas' work, which is reassuring. We thank him for this comment and report this alignment by citing Coltice et al. (2023).

For our Taylor tests, we utilise the pyadjoint library's implementation, available at pyadjoint verification. The default setting for halving the perturbation is set to 4 consequent divisions, which spans $\approx$ two orders of residuals. To ensure reproducibility, we maintain the Taylor tests in our figures but confirm the results presented in our response in the manuscript. This way we maintain the standard form of Taylor tests in our paper, aligning with Dolfin-Adjoint upstream repository.

N. C.–C3: *The automatic differentiation section (Dolfin). This is very important to be more explicit since it is a new methodology that you take to geodynamics, and this is precisely this methodology that empower your workflow. It is too abstract now. I would prefer you explain more precisely how it works and how it respects the idea that the generated adjoint is the exact adjoint of the forward code. I understand it is hard, since it is why I have such hard time publishing my preprint. But a little more would be nice. Maybe I did not have enough time, but it seems to me that some of the important preparation work to make for automatic differentiation is not explicit. Do you have directives in the forward code? Where do you tape, or checkpoint? What is your organization? It is nice to give a glimpse to the future users of what to be careful about if they want to modify what you have done? Off course, if it's too technical in the paper, that may not work.*

Response – 3: We recognise the challenge in elucidating the concept of automatic differentiation (AD) for the geodynamic community. The essence of our framework's AD capability, particularly through Dolfin-Adjoint, can be appreciated by understanding several key points. We have now added details of how Dolfin-Adjoint strategy in taping and generating blocks to Sec. 2.2.2. The brief overview aims to introduce our approach succinctly and a more detailed technical exposition might be beyond the geodynamic community's general interest. Our primary objective in developing G-ADOPT has been to decouple these technical aspects from the user community's concerns, thereby promoting the broader application of adjoint inverse methodologies in geodynamics.

N. C.–C4: *I would put the derivation of the adjoint equation in an Appendix, you could refer to when you talk about the linear properties of the adjoint with pseudo-plasticity.*

Response – 4: We appreciate the suggestion to relocate the derivation of the adjoint equation to an appendix to enhance the readability of our manuscript. Our initial decision to incorporate this derivation within the main text was informed by the guidelines provided by GMD, which advocate for including "all material required to understand the essential aspects of the paper, such as experimental methods, data, and interpretation, should preferably be included in the main text." Considering the brevity of the derivation, we have opted to retain it in the main text. However, we are open to

[Figure]

Figure 1: Expansion of the Taylor test for the two scenarios in our study, top: two dimensional annulus domain with non-linear rheology, bottom: two dimensional rectangular case with constant viscosity. In all cases the convergence behaviour of $O(h^{-2})$ approaches a machine precision, except for the $T_F$ and $u_S$ misfits in the annulus case. In those cases the accuracy of the derivative is controled by the solver tolerance set for SNES. Our tests confirms the behaviour shown previously in Coltice et al. (2023).

relocating the derivation to the Appendix should the editor recommend this adjustment, as our primary adherence is to the guidelines set forth by GMD.

**N. C.–C5:** *In my opinion, the discussion needs reworking. For now, most of it reads like a conclusion. The results should be compared to former studies. Bocher et al. (2016, 2018) are ideal because the models have similarities in geometry and objectives. You can discuss then, how the inversion performs. You have all the results for this in your figures. You have to discuss also computing time. This is very important also because you state you want your method to tackle grand challenges of 3D spherical reconstruction with self-generating plates. For now your forward model is 2D, with only 3 orders of magnitude variations in viscosity, which is not enough yet to have plate-like behavior. What is the computing power needed for inversion of this 2D case? To move on, how do you envision going to 6 or more orders of magnitude and 3D? How many million years will you be able to invert for given your computing time? This is a necessary discussion here, since you bring the ambition of the method from the start.*

**Response – 5:** The primary objective of this manuscript is to introduce our computational platform to the community and validate its effectiveness, building upon insights from preceding adjoint studies. It was not within our scope to assert the superiority of the adjoint method over other methods, such as the Ensemble Kalman Filter, or to assess method efficiency in Earth-like scenarios. Consequently, we do not see a motivation or need to prioritise comparisons with studies in similar geometries, including those by Bocher et al. (2016). However, addressing the reviewer's specific points:

(i) Comprehensive timing data for the simulations featured in this study, as well as in our previous work (Davies et al., 2022), are regularly updated and made available through our GitHub actions, an example of which is accessible via the provided link.

(ii) We anticipate that increasing the viscosity contrast to up to 6 orders of magnitude will be feasible without significant issues. Our computational solvers and preconditioners are designed to handle such variations in viscosity contrast efficiently, as demonstrated in Davies et al. (2022).

(iii) Transitioning to 3-D simulations will undoubtedly require more computational resources. We have already validated our methodology in 3-D settings and, through access to the National Computational Infrastructure (NCI) in Australia, we are well-positioned regarding computational capacity. Our research group is among the top 10 CPU time allocates across all science and engineering disciplines in the country, ensuring the tractability of these more complex simulations.

In summary, while the manuscript's focus was not on direct methodological comparisons or detailed computational logistics, we recognise the value of these considerations and have thus provided responses to the specific inquiries raised by the reviewer.

**N. C.–C6:** *Recommendations along the text:*
*Davies and Richards, 1992 in the intro: you should propose a more up to date review here, since so many discoveries have been made since.*

**Response – 6:** Our initial citation of Davies and Richards (1992) in the introduction was intended to underscore the well-established nature of the theory under discussion. Acknowledging the reviewer's recommendation for a more contemporary review, we have now included a citation to Coltice et al. (2017), offering readers access to more recent insights and developments.

**N. C.–C7:** *l.36: « state is unknown »: not only. Also because the physics we use is incomplete, and the physical and chemical properties of the Earth remain uncertain.*

**Response – 7:** We have now changed the sentence to reflect this.

**N. C.–C8:** *l.39: Rolf and Tackley citation here does not feel appropriate. If you want to add a citation with pseudo-plasticity, you souhld choose one focusing on the equations.*

**Response – 8:** We have now removed the citation to Rolf and Tackley and added Zhong et al. (2007).

**N. C.–C9:** *l.43: all the citations here are from the same group, except Dannberg, although other groups have used state-of-the-art 3D convection calculations at high resolution with different codes. Actually, you cite some of them in the next paragraph so you*

*could cite them here too.*
Response – 9: We have updated the citations as suggested.

N. C.–C10: *l.44: I would remove the discussion on Monte-Carlo technique. It's not necessary here. It feels out of the scope.*
Response – 10: We concur with the reviewer that the mention of Monte-Carlo techniques may not be strictly essential to our introduction. However, we believe its inclusion, albeit not significantly extending the length of the introduction, underscores a critical aspect of the cost associated with addressing our problem, which we return to in our discussion.

N. C.–C11: *l.50: Bocher et al. (2016) is a Kalman filter. So it should be cited later. Bocher et al. (2018) is ensemble Kalman filter. Both these methods are doing a similar work as you propose later so they are useful for you to compare. The major difference is that we did not use exactly the same « observations » for the inversion and did apply a Kalman smoother to correct past states, which is something you are interested in.*
Response – 11: We agree with the reviewer's observation and have accordingly revised the citations to accurately reflect the distinction between the methodologies employed in Bocher et al. (2016) and Bocher et al. (2018).

N. C.–C12: *l.53. "assimilated" is not a correct word here. I know it has been misused for years. But now you are considering REAL assimilation of surface kinematics, you see the problem. What you call assimilation here is forcing the surface of a model, so it is a forward operation. While assimilation in data assimilation is an inverse operation, specifically the one you are designing here. So please, change it to « nudged » or some more appropriate term.*
Response – 12: We have now changed the text to reflect reviewer's concerns.

N. C.–C13: *Figure 1. In line with the « assimilation » word. Case (a) can be a « nudged » model. It is a way to differentiate nudging (a), KF data assimilation (b) and 4Dvar data assimilation (c).*
Response – 13: Acknowledging the distinction highlighted by the reviewer between "nudging" and KF data assimilation methods, we have now added these terms in the caption to highlight this issue.

N. C.–C14: *L.62: Kalman smoother is a Kalman filter method to do retro-propagate the information in past states. So it is possible to use Kalman filter and improve the knowledge on initial conditions. The major difference with the adjoint is that it is a statistical treatment whereas the adjoint keeps the physical consistency (it has its advantages and disadvantages, considering that the physical model is not perfect).*
Response – 14: Kalman filtering, as depicted in Figure 1 of our manuscript, is characterised by its forward temporal correction of state variables upon the acquisition of new observations. Kalman smoothing, on the other hand, refines this process by utilising all available data, irrespective of their temporal positioning, to update state variables at any given time. This characteristic has been identified by Bocher et al. (2016) as an extension to their approach for integrating seismic tomography data.

This argument overlooks a critical aspect when applied to advection-diffusion problems, such as mantle convection. The back-propagation of information in time necessitates the construction of a sensitivity model. In the realm of mantle convection simulations, this is invariably accomplished through the development of an adjoint model. Hence, instances of Kalman smoothing in Earth system modelling typically require the formulation of both tangent linear and adjoint models to facilitate the bi-directional transmission of sensitivity information through time.

It is, therefore, not entirely accurate to suggest Kalman smoothing as an alternative to adjoint modeling within the context of mantle convection. Instead, Kalman smoothing in such settings effectively relies on employing adjoint models. Consequently, our manuscript's focus on optimisation based reconstructions of mantle convection over time using the adjoint method does not preclude the utility of our library for implementing Kalman smoothing approaches. Moreover, our assertion that sequential filtering predominantly propagates information forward in time, remains valid. Without an adjoint model, backward information propagation to refine past state estimates based on future observations is not feasible. Thus, our original statement stands as both accurate and relevant within the scope of our discussion.

N. C.–C15: *l.65: I would remove that idea of an emerging field. We could say that the inverse models of the geoid back in 1990s were inverse geodynamics. Or the pioneering works of Peter and Alik are 20 years old already and we cannot say the technique has penetrated the geodynamics community. Actually, what you propose could help to do so.*
Response – 15: We changed 'rapidly emerging' to 'rapidly evolving'.

N. C.–C16: *l.74. About Li et al. (2017). It can be important to be nuanced saying that such recovery is limited by the ill-posedness,*

*which is their conclusion.*
Response – 16:  We now complete this sentence by mentioning the ill-possedness of their problem.

N. C.–C17:  *Intro: I would say in the intro that solving adjoint equations does not necessarily generate the exact adjoint of the forward code, while AD does. Hence, your approach enforces that the adjoint code is the exact adjoint of the forward code, which is necessary for optimal inversion.*
Response – 17:  In response to this comment we have added the words "fully consistent forward and adjoint" towards the end of the intro, to emphasise this distinction more early on and whet the reader's appetite.

N. C.–C18:  *Table 1: Delta T = Temperature difference between top and bottom ; d= thickness of the system. It is not just characteristic something.*
Response – 18:  These are now corrected.

N. C.–C19:  *l.219: it would be nice to cite Talagrand, 1997 or some of his pioneering work here.*
Response – 19:  We have now added the reference to Talagrand (1997).

N. C.–C20:  *l.225-230: Okay for the advantages. But which drawbacks? –> not the exact adjoint of the forward code which may not be good for inversions. Write a new code for each new problem (set of equations).*
Response – 20:  We fully acknowledge Nicolas' remarks. We believe these concerns are addressed in subsequent paragraphs and within our discussion section. This response may have preceded a review of those sections, where we elaborate on these limitations, which indeed partly motivated our approach.

N. C.–C21:  *l.231: I think here you could start with AD right away because with AD you find a solution to both the limitations of the continuous adjoint: exact adjoint and versatility of problems (not rewriting the code if you change the problem/update the code). I would reduce the length and you go straight to the point.*
Response – 21:  The discussion on continuous versus discrete adjoint methods remains a significant debate within the optimisation community, which we consider crucial to the discipline. This manuscript engages with this debate, notably reflecting on insights from Georg Reuber, who underscored the merits of the continuous (or manually derived) adjoint approach. Our strategy aims to synthesise the strengths of both approaches, utilising the abstract representation of mathematical operations in Firedrake to harness the respective benefits. Nonetheless, we argue that a thorough grasp of automatic differentiation (AD) requires preceding discussion on the intricacies of continuous adjoint methods. This foundation is essential for delineating how AD addresses these inherent challenges. We also wish to highlight that the efficiency of AD-generated adjoint codes remains a pivotal concern for AD methodologies. Firedrake's use of abstract mathematical representations is instrumental in achieving optimal efficiency in our derivative calculations.

N. C.–C22:  *l.233: you can cite Giering and Kaminski, 1998 for AD pioneering work.*
Response – 22:  We have now cited Giering and Kaminski (1998)

N. C.–C23:  *l.240: too bad you forget to cite some existing AD tools used for science application purposes: Tapenade, TAF, or Enzyme.*
Response – 23:  We recognise that the array of existing AD tools applicable to scientific problems is extensive and not confined to those highlighted by Nicolas. Given the finite number of citations permissible within our manuscript – which does not aim to serve as an exhaustive review of AD tools – we have selectively cited key well known examples. Alongside the previously cited tools in Abadi et al. (2015) and Paszke et al. (2019), we have now incorporated a citation for Enzyme software through Moses et al. (2022), broadening our discussion to include this additional tool.

N. C.–C24:  *l. 240 and equivalence of the methods: I don't agree: the solvers, the methods involve a suite of operations that are discrete anyways and can involve some differences that can possibly prevent the continuous adjoint to be the exact adjoint. It's not only a question of resolution but the approximations you make also, and the tools you use to solve the problem.*
Response – 24:  We agree that the potential for discrepancies is indeed higher in the continuous adjoint scenario. However, it is feasible to utilise temporal and spatial discretisations that are consistent (at least theoretically, there should exist such a configuration) to ensure alignment between the forward and adjoint codes. Our reference to infinite spatial and temporal resolutions aims to underscore a hypothetical scenario where continuous and discrete representations of PDEs converge. Clearly, such a scenario is not achievable in practice but serves to highlight a theoretical consideration.

This statement draws upon the insightful analysis provided by Giles and Pierce (2000), particularly their examination of both approaches on page 408.

N. C.–C25: *2.2.2 -> More would be needed here. It's a major point of your work.*
Response – 25: We have addressed this comment in the major comments above.

N. C.–C26: *l.560: I find your forward model to be a little oversold here. With 3 orders of magnitude viscosity change, you don't make plates in convection models, and it's even more unlikely to have Earth-like subduction zones (one-sided etc...). Using this model is perfect for the demo here, but I am sure you want to have a more plate-like behaviour in the mode in the future, which means necessarily increasing this contrast.*
Response – 26: We have now removed the term Earth-like.

N. C.–C27: *For the duration of your model, it is worth giving a glimpse of what Earth time it corresponds to, so that we can evaluate how far back in time we can go given the uncertainty on the past state we look for. Simply putting dimensions here it would be close to 200Myrs. But probably the convection velocities are lower when I look at your figure and it would probably translate to half of this maybe? That would be useful.*
Response – 27: The reviewer's analysis regarding the duration of our simulations is helpful. Upon dimensionalising the time in our later simulations, it equates to approximately 250 Myrs. However, considering the distances traversed by various features within these simulations, a more realistic representation would be less than 100 Myrs. We have incorporated this estimate into our result section.

N. C.–C28: *l.612: Visco-plasticity means surface decoupling and more non-linearities. It matters and you could show it by showing the adjoint: the sensitivity of your misfit to surface velocities is probably lower than the isoviscous case. But again, 3 orders of magnitude is not really plate-like so pushing the interpretation is not necessarily good here.*
Response – 28: We agree that the visco-plastic behaviour in our models is not strong and likely does not account for the observed reduction in misfit. Hence, attributing decreased misfit reduction solely to visco-plasticity may be misleading in this context. We have now removed the emphasis on the impact of nonlinearities.

N. C.–C29: *l.663: surface velocities, except today, are not observed. They come from kinematic models.*
Response – 29: This concern essentially pertains to semantic distinctions, reflecting the terminology commonly utilised within our community. Similarly, one might contend that even present-day observed plate velocities essentially constitute a model solution inverted to optimally represent temporal variations captured by GNSS data globally. For the purposes of inverse problems as examined in our study, "observations" refer to constraints derived from external disciplines. It is acknowledged that the uncertainty inherent in these models may vary significantly, particularly with older plate reconstruction models.

N. C.–C30: *708: Seton et al. (2023) provides an overview of plate reconstructions that should be discussed here.*
Response – 30: We have incorporated a citation to Seton et al. (2023), acknowledging its comprehensive overview of plate reconstructions.

N. C.–C31: *711: It's not an inversion, but the work of Bello et al. (2015) assesses in some ways the errors produced on the forward modelling.*
Response – 31: We acknowledge the insightful contribution of Bello et al. (2015) to the understanding of error propagation in forward modelling of mantle convection. However, the focus in the referenced section of our manuscript is on the implications of uncertainty in the inverse modelling of mantle flow. In this context, the study by Colli et al. (2020) provides a more relevant discussion and thus is the appropriate citation for our purposes.

N. C.–C32: *715: The issue with smooth solution is that geodynamic features are sharp and their shape is fundamental for the self-consistency of the flow, especially with variable viscosity. This is very different from seismic or gravity inversions in which smoothing does not hinders the fundamentals of the physics at play.*
Response – 32: The reviewer's point is well taken. The self-consistent development of many geodynamic features critically depends on the sharpness of these features. To mitigate issues stemming from the inherent uncertainty in seismic imaging of the mantle, our approach incorporates a regularisation term associated with Total Variation Regularisation.

It is crucial to clarify that we do not suggest past mantle features should be inherently smooth. Rather, our methodology entails imposing a degree of smoothness on the solutions to prevent overfitting, despite this introducing a level of

smoothness that may not align with the precise 'physics at play'. This aspect of regularisation becomes increasingly significant when dealing with uncertain observations, presenting a trade-off that, in our view, constitutes the lesser of two evils.

N. C.–C33: *l.723: not "assimilation", what you do now with surface velocities is real assimilation.*
Response – 33: Corrected.

N. C.–C34: *Conclusion can be reduced to the essentials.*
Response – 34: We have now reduced the text significantly.

**References cited**

Abadi, M., Agarwal, A., Barham, P., Brevdo, E., Chen, Z., Citro, C., Corrado, G. S., Davis, A., Dean, J., Devin, M., Ghemawat, S., Goodfellow, I., Harp, A., Irving, G., Isard, M., Jia, Y., Jozefowicz, R., Kaiser, L., Kudlur, M., Levenberg, J., Mané, D., Monga, R., Moore, S., Murray, D., Olah, C., Schuster, M., Shlens, J., Steiner, B., Sutskever, I., Talwar, K., Tucker, P., Vanhoucke, V., Vasudevan, V., Viégas, F., Vinyals, O., Warden, P., Wattenberg, M., Wicke, M., Yu, Y., and Zheng, X.: TensorFlow: Large-Scale Machine Learning on Heterogeneous Systems, URL https://www.tensorflow.org/, software available from tensorflow.org, 2015.

Bello, L., Coltice, N., Rolf, T., and Tackley, P. J.: On the predictability limit of convection models of the Earth's mantle, Geochemistry, Geophysics, Geosystems, 15, 2319–2328, https://doi.org/10.1002/2014GC005254, 2014.

Bello, L., Coltice, N., Tackley, P. J., Dietmar Mueller, R., and Cannon, J.: Assessing the role of slab rheology in coupled plate-mantle convection models, Earth and Planetary Science Letters, 430, 191–201, https://doi.org/10.1016/j.epsl.2015.08.010, 2015.

[revised manuscript text omitted]

---

## Author Response (AR2)

Second Round of Responses to Reviewer Comments for:

**Automatic adjoint-based inversion schemes for geodynamics: Reconstructing the evolution of Earth's mantle in space and time**

Sia Ghelichkhan, Angus Gibson, D. Rhodri Davies, Stephan C. Kramer, and David A. Ham

May 6, 2024

We again thank the reviewers, Georg Reuber, Nicolas Coltice for reading our reponses to their reviews. We also thanks to the editorial team for excellent handling of the review process. Below is our response to the reminaing comment from Nicolas marked with "Response - [#] ".

**Responses to Reviewer 3: (Nicolas Coltice)**

N. C.–C1: *From my assessment, the authors have answered the questions and proposed appropriate modifications. There are two exceptions, being minor revisions, that I suggest the authors do: - replace Fig.4 and Fig.9 that do not prove their point by Fig.1 in the answer to the review that prove their point. This Fig.1 shows the consistency of their approach without a doubt and should be now within the manuscript. Little text to go with them is needed. - Fig.10 shows peculiar optimisation curves, with changes of states. Theses increase in cost function while optimizing have to be discussed within the text, which is not the case yet. With both these changes, the manuscript is good to go as far as I am concerned. It shows great progress in building consistent reconstruction of mantle circulation.*

Response – 1: We sincerely thank Nicolas for his detailed and insightful feedback, which has significantly strengthened our manuscript. In response to the comments:

1. We have replaced Figures 4 and 9 with Figure 1 from our responses, as suggested. The new figures effectively illustrates the consistency of our approach, reinforcing the main points of our discussion.

2. We have revised the optimisation curves in Figure 10. The noted issues were due to incorrect sorting of output file names as a result of change in a sorting function, which has now been rectified.